# Integral equation solutions for the average run length for monitoring shifts in the mean of a generalized seasonal ARFIMAX($P$, $D$, $Q$, $r$)$_s$ process running on a CUSUM control chart

**Yupaporn Areepong, Wilasinee Peerajit***

Department of Applied Statistics, Faculty of Applied Science, King Mongkut's University of Technology North Bangkok, Bangkok, Thailand

* wilasinee.p@sci.kmutnb.ac.th

**Data Availability Statement:** All relevant data are within the paper and its Supporting information files.

## Abstract

The CUSUM control chart is suitable for detecting small to moderate parameter shifts for processes involving autocorrelated data. The average run length (ARL) can be used to assess the ability of a CUSUM control chart to detect changes in a long-memory seasonal autoregressive fractionally integrated moving average with exogenous variable (SARFIMAX) process with underlying exponential white noise. Herein, new ARLs via an analytical integral equation (IE) solution as an analytical IE and a numerical IE method to test a CUSUM control chart's ability to detect a wide range of shifts in the mean of a SARFIMAX ($P$, $D$, $Q$, $r$)$_s$ process with underlying exponential white noise are presented. The analytical IE formulas were derived by using the Fredholm integral equation of the second type while the numerical IE method for the approximate ARL is based on quadrature rules. After applying Banach's fixed-point theorem to guarantee its existence and uniqueness, the precision of the proposed analytical IE ARL was the same as the numerical IE method. The sensitivity and accuracy of the ARLs based on both methods were assessed on a CUSUM control chart running a SARFIMAX($P$, $D$, $Q$, $r$)$_s$ process with underlying exponential white noise. The results of an extensive numerical study comprising the examination of a wide variety of out-of-control situations and computational schemes reveal that none of the methods outperformed the IE. Specifically, the computational scheme is easier and can be completed in one step. Hence, it is recommended for use in this situation. An illustrative example based on real data is also provided, the results of which were found to be in accordance with the research results.

## Introduction

The discipline of statistical process control (SPC) provides tools such as control charts for monitoring processes and detecting changes in a given in-control model. The Shewhart, cumulative sum (CUSUM), and exponentially weighted moving average (EWMA) control

**Funding:** This work was supported by Grant No. KMUTNBBasicR-64-20, National Science, Research and Innovation Fund (NSRF) via King Mongkut's University of Technology North Bangkok, to Wilasinee Peerajit. The funders had no role in study design, data collection and analysis, decision to publish, or preparation of the manuscript.

**Competing interests:** The authors have declared that no competing interests exist.

charts are the three most commonly used ones. The CUSUM and EWMA control charts can effectively detect small shifts in the parameters of interest (usually location parameters) whereas the Shewhart control chart such as the well-known $\bar{X}$-chart is used to detect large shifts in process parameters. Details of the charts can be found in Montgomery [1]. The CUSUM control chart was used in the present study due to its superior capability of detecting small parameter shifts and to take advantage of the benefits of using an upper-sided control chart.

Small levels of autocorrelation between successive observations can have a significant effect on the statistical properties of control charts. Numerous researchers have considered the effect of autocorrelation on control chart performance. Johnson and Bagshaw [2] and Bagshaw and Johnson [3] concluded that even small levels of autocorrelation generated by using autoregressive AR(1) or moving average MA(1) time series models made it more difficult to detect out-of-control signals on a conventional CUSUM control chart. Nevertheless, a CUSUM control chart has been used for monitoring shifts in the mean of a stationary ARMA (*p*, *q*) model process, and its performance has been evaluated when applied to processes with autocorrelated data [4].

Observations from real-world stochastic processes frequently exhibit a time series component. In particular, economic observations can have AR and MA components in a time series model. The error or so-called white noise, which is defined as the difference between the actual and approximated values, should be kept to a minimum to maintain the highest possible accuracy rate. Although the white noise (also referred to as Gaussian white noise) generated by autocorrelated observations usually follows a normal distribution, this is not always the case. Indeed, non-Gaussian white noise processes are useful for modeling a wide variety of phenomena, including wind speed, and the oxygen concentration and flow rate of water. Numerous studies have been conducted on time series models with non-Gaussian white noise, and when it is exponentially distributed (referred to as exponential white noise) is particularly interesting. Jacob and Lewis [5] considered an ARMA(1,1) process with exponential white noise, while Mohamed and Hocine [6] conducted a Bayesian analysis of an AR(1) process with exponential white noise. Furthermore, Pereira and Turkman [7] conducted a Bayesian analysis of threshold AR models with exponential white noise, while Suparman [8] recently proposed parameter estimations for an AR model of unknown order with exponential white noise.

Time series models are capable of capturing process correlation. An important class of time series model is the stationary process in which it is assumed that the process remains stable around a constant mean. For monitoring autocorrelated processes, the type of model can provide a framework for establishing statistical control. The following basic time series models were used in this study. The conventional Box-Jenkins AR integrated MA (ARIMA) model can be generalized as the AR fractionally integrated MA (ARFIMA) model that allows non-integer (fractional) values for the differencing parameter. This is advantageous when modeling time series with inherent long-range dependence (i.e., long-memory) [9, 10]. Accordingly, ARFIMA models are appropriate for time series long-memory processes, and exogenous variables can be incorporated into the time series model (i.e., ARFIMAX) to improve the latter's performance. Ebens [11] first introduced the ARFIMAX model with *k* exogenous variables and used it to estimate the realized volatilities in a Dow Jones Industrial Average portfolio, while Degiannakis [12] estimated and forecasted intraday realized volatility using ARFIMAX and ARFIMAX-TARCH (threshold AR conditional heteroskedasticity) models.

Seasonality (seasonal periods over a single year) is a phenomenon that affects time series. Numerous scholars have discussed ARFIMA models with seasonal components, and it has been used to explain a large number of real-world cyclical phenomena in long-memory

processes. For instance, it occurs in series of revenues, inflation rates, monetary aggregates, and gross national product. In the current study, we concentrate on seasonal ARFIMAX (SARFIMAX) models. In terms of economic forecasting and other fields, cases where an exogenous variable is included in the forecasting model are usually more accurate than ones without it. In addition, time series model processes have been monitored by using control charts. For instance, Ramjee [13] discovered that the performances of Shewhart and EWMA control charts with correlated observations in an ARFIMA model were not good, and so proposed the hyperbolic weighted MA (HWMA) control chart instead. Subsequently, Ramjee et al. [14] designed an HWMA forecast-based control chart particularly designed for a non-stationary ARFIMA model with autocorrelated data. When residual control charts with ARFIMA and ARIMA models were used to monitor the air quality Taiwan [15], the one with the ARFIMA model was more appropriate than the one with the ARIMA model. Rabyk and Schmid [16] recently introduced the EWMA control chart for detecting changes in the mean of a long-memory process with the control chart's design based on an ARFIMA(*p*, *d*, *q*) process.

The most widespread measure of evaluating the performance of the control chart is the average run length (ARL). The power of a control chart is defined as the probability of its out-of-control signal detection, while the average number of samples required to signal an out-of-control situation in the process (or expected value of the run length) is represented by the ARL. The in-control and out-of-control ARLs are denoted as $ARL_0$ and $ARL_1$, respectively: $ARL_0$ should be as large as possible whereas $ARL_1$ should be as small as possible when the process changes undesirably. To improve the out-of-control detection performance on a CUSUM control chart, two alternative schemes are proposed that help to minimize $ARL_1$ when $ARL_0$ is fixed.

Assessing the ARL of a control chart using randomness and numerical methods is possible using a variety of techniques, including Monte Carlo, Markov Chain, and integral equations (IEs). The Monte Carlo simulation technique can be used to determine the expected run length until the target ARL value approaches that of an IE under the assumption of a sufficiently long simulation time. For the Markov Chain approach, the range between the upper and lower control limits is divided into sub-intervals, while the probabilities of the runs on the control chart are presented as a transitional probability matrix [17] over the sub-interval so that the ARL can be evaluated by inverting the matrix. In the meantime, the IE method comprises two alternatives: analytical IE and numerical IE. The analytical IE method generates exact ARLs or explicit formulas. The numerical IE method is transformed into a system of linear equations that can be solved for the ARL by using certain linear algebra-based methods. Crowder [18] provided a numerical method by using an IE of the second kind and a computer program for ARL evaluation of a normally distributed process on a two-sided EWMA control chart. Moreover, Hawkins [19] solved IEs for determining the ARL of a CUSUM control chart. Later, ARLs by using analytical IEs and the Markov Chain approach were used on EWMA and CUSUM control charts for observations from an AR(1) process with additional random error [20]. To evaluate the ARL for a SARMA(1, 1)$_s$ process with exponential white noise on a CUSUM control chart, Phanyaem [21] developed explicit formulas for IEs initially based on the SARMA model. In addition, analytical IEs for the solution for the ARL for a long-memory SARFIMA model on CUSUM control chart were presented by Peerajit et al. [22]. Recently, complex data from seasonal and non-seasonal MA processes with exogenous variables were used to evaluate the ARL on a CUSUM control chart by Sunthornwat and Areepong [23]. Moreover, it is important to apply Banach's fixed-point theorem to prove the existence and uniqueness of an ARL. In this paper, our focus was on establishing ARLs for monitoring changes in a long-memory SARFIMAX process with underlying exponential white noise running on a CUSUM control chart.

The remainder of the article is organized as follows. Section 2 contains a brief review of the ARFIMAX models taking long memory and seasonality into account and descriptions of the CUSUM control chart, as well as the characteristics of the ARL and Banach's fixed-point theorem. Theoretical proof for the existence and uniqueness of analytical IE, along with its comparable accuracy to the numerical ARL are covered in Section 3. The next section comprises derivations of the proposed analytical and numerical IE methods for the ARL for monitoring changes in a long-memory SARFIMAX process with underlying exponential white noise on a CUSUM control chart. The results of an evaluation of the performances of the IE methods are provided in Section 5. In Section 6, application of the methods to the monthly stock price data for PTT Public Company Ltd with the EUR/THB exchange rate as the exogenous variable is reported. Conclusions and a discussion of possible future work are provided in Section 7. Finally, the appendix contains technical details of the computation of the ARL by using the proposed methods.

## Preliminaries

The relevant fundamentals of the long-memory SARFIMAX process with underlying exponential white noise are described in this section. In addition, we determine whether detecting changes in the mean of a generalized SARFIMAX process running on a CUSUM control chart is suitable for this study. For simplicity, the second subsection is focused on the upper-sided CUSUM control chart only. Next, performance evaluation of a control chart and the characteristics of the ARL are covered. In the last subsection, definitions related to Banach's fixed-point theorem are covered.

### Long-memory SARFIMAX processes with underlying exponential white noise

The ARFIMA model, a generalized form of the usual Box-Jenkins ARIMA model but allowing non-integer (fractional) values for the differencing parameter, is useful for modeling time series with inherent long-range dependence [9, 10]. Furthermore, including an exogenous variable in the ARFIMA model produces the ARFIMAX model with better performance [12]. In practice, time series are often found in meteorology, economics, hydrology, and astronomy [24]. Seasonality is an autocorrelation structure where the data vary according to specific periods in a year, a phenomenon that is included in our analysis.

A SARFIMAX model of order $(P, D, Q, r)_s$, with AR order ($P$), fractional integration order ($D$), MA order ($Q$), and exogenous variables order ($r$), and period ($s$) for seasonal data is an extension of the ARFIMAX($p, d, q, r$) model applied to a seasonal time series. SARFIMAX($P, D, Q, r$)$_s$ is stationary and invertible for $D \in (-1, 0.5)$; for $D > 0$, where $D$ is seasonal fractionally differentiated/ integration and $D \in (0, 0.5)$ is for a long-memory process on a CUSUM control chart.

**Definition 1**. Let $\{Y_t\}_{t \in \mathbb{Z}}$ be a sequence from a SARFIMAX($P, D, Q, r$)$_s$ process defined as

$$\Phi_P(B^s)(1 - B^s)^D Y_t \;=\; \mu + \Theta_Q(B^s)\xi_t + \sum_{l=1}^{r}\omega_l X_{lt}, \;\; \xi_t \sim Exp(\beta). \tag{1}$$

where $\mu$ is the constant of the process, $X_{lt}, l = 1, 2, \ldots, r$ are exogenous variables at time $t$, The $\omega_l, l = 1, 2, \ldots, r$ are coefficients corresponding to $r$ exogenous variables. It is assumed that the error terms ($\xi_t$) are white noise, i.e., $\xi_t \sim Exp(\beta)$. The operator $B$ is a backward-shift operator (i.e., $B^{sk}Y_t = Y_{t-sk}$), $(1 - B^s)^D$ is seasonal fractional difference operator, where $s$ refers to the number of seasonal periods per year (e.g., $s = 12$ for monthly data), and $\Phi_P(B^s)$ and $\Theta_Q(B^s)$ are

the coefficient of seasonal AR and MA polynomials in *B* order *P* and *Q*; $s \in \mathbb{N}$, respectively, defined by

$$\Phi_P(B^s) = 1 - \Phi_1 B^s - \Phi_2 B^{2s} - \ldots - \Phi_P B^{Ps} = 1 - \sum_{i=1}^{P} \Phi_i B^{is}$$

and

$$\Theta_Q(B^s) = 1 - \Theta_1 B^s - \Theta_2 B^{2s} - \ldots - \Theta_P B^{Qs} = 1 - \sum_{j=1}^{Q} \Theta_j B^{js},$$

where the coefficient of seasonal $\Phi_i$, $i = 1.2, \ldots, P$, and $\Theta_j$, $j = 1, 2, \ldots, Q$ are real numbers.

**Remark**: For all real *D*, seasonal fractional difference operator $(1 - B^s)^D$ with seasonality $s \in \mathbb{N}$ is defined by employing a binomial expansion as follows:

$$(1 - B^s)^D = \sum_{k \geq 0} \binom{D}{k} (-1)^k B^{ks} = 1 - DB^s + \frac{D(D-1)}{2!} B^{2s} - \frac{D(D-1)(D-2)}{3!} B^{2s} + \ldots ,$$

where *D* becomes crucial for describing the degree of seasonal persistence.

The result of the process is based on binomial expansion. After simplification of the derivation of process, the generalized SARFIMAX(*P*, *D*, *Q*, *r*)$_s$ process is as follows:

$$
\begin{aligned}
Y_t =\ & \mu + \xi_t - \sum_{j=1}^{Q} \Theta_j \xi_{t-js} + \sum_{l=1}^{r} \omega_l X_{lt} + \left( DY_{t-s} - \frac{D(D-1)}{2!} Y_{t-2s} + \frac{D(D-1)(D-2)}{3!} Y_{t-3s} - \ldots \right) \\
& + \sum_{i=1}^{P} \Phi_i \left( Y_{t-is} - DY_{t-(i+1)s} + \frac{D(D-1)}{2!} Y_{t-(i+2)s} - \frac{D(D-1)(D-2)}{3!} Y_{t-(i+3)s} + \ldots \right),
\end{aligned}
\tag{2}
$$

where initial value is normally the process mean, seasonal AR, seasonal MA and exogenous variables coefficient are, respectively, $\xi_{t-s}, \xi_{t-2s}, \ldots, \xi_{t-Qs} = 1, -1 \leq \Phi_i \leq 1, -1 \leq \Theta_j \leq 1$, and $-1 \leq \omega_l \leq 1$. It is assumed that the initial value of generalized SARFIMAX(*P*, *D*, *Q*, *r*)$_s$ process $Y_{t-s}, Y_{t-2s}, \ldots, Y_{t-Ps}, Y_{t-(P+1)s}, \ldots = 1$. In this paper, $Y_t$ is long memory process which is assess $D \in (0, 0.5)$.

**Process monitoring on a CUSUM and EWMA control charts.** One of the simplest and efficient SPC techniques for detecting small and moderate changes in the mean of a process is the CUSUM and EWMA control charts. The statistic of the upper-sided CUSUM control chart [25] can be expressed by the following recursive equation:

$$C_t = \max \{0,\ C_{t-1} + Y_t - \eta\}, \text{ for } t = 1,\ 2, \ldots ,
\tag{3}$$

where CUSUM control chart parameter $Y_t$ is a sequence of long-memory SARFIMAX(*P*, *D*, *Q*, *r*)$_s$ process with underlying exponential white noise. $\eta > 0$ and $C_0 \geq 0$ as the reference and starting value, respectively. The starting value of the CUSUM statistic is $\varphi$ (i.e., $C_0 = \varphi$, $\varphi$ is the initial value; $\varphi \in [0, b]$), and parameter $\eta$ is used to balance the series ($C_t$) that is commonly chosen to be larger than but close to the value of the in-control mean.

**Remark**: If $C_t$ control limit $b > 0$, an alarm is triggered that indicates that the process might be out-of-control. In view of an exact performance evaluation, we chose rational or even real number design parameter $\eta$.

For the above CUSUM sequence ($C_t$), the corresponding stopping time with predetermined threshold *b* can be deployed:

$$\tau_b = \inf \ \{t > 0 \ ; \ C_t > b\}, \ \text{for} \ \varphi \leq b, \tag{4}$$

where $\tau_b$ is the stopping time, *b* is the upper control limit (UCL) of CUSUM chart.

The statistic of the upper-sided EWMA control chart [26] can be expressed by the following recursive equation:

$$Z_t = (1 - \lambda)Z_{t-1} + \lambda Y_t, \ \text{for} \ t = 1, \ 2, \dots \ ,$$

where $\lambda$ is an exponential smoothing parameter with $0 < \lambda \leq 1$. The starting value of the EWMA statistic is $Z_0 = Y_0$, the target mean value is $\mu$, $Y_t$ is a processes with mean $\mu$ and variance $\sigma^2$.

The upper control limit (UCL) and lower control limit (LCL) to detect the sequence is given by,

$$
\begin{aligned}
UCL &= \mu_0 + L\sigma\sqrt{\frac{\lambda}{2 - \lambda}[1 - (1 - \lambda)^{2t}]} \ , \\
LCL &= \mu_0 - L\sigma\sqrt{\frac{\lambda}{2 - \lambda}[1 - (1 - \lambda)^{2t}]} \ ,
\end{aligned}
\tag{5}
$$

where *L* is the width of the control limits, where $\mu_0$ is the target mean, $\sigma$ is the process standard deviation.

**Characteristics of the ARL.**   Throughout the whole paper, the error term ($\xi_t$) is a sequence of continuous i.i.d. random variables taken from an exponential distribution with distribution function. ($F(x, \beta)$), which is used in the next section concerning monitoring of the mean. We consider the following simplified change-point models:

$$
\xi_t \sim
\begin{cases}
Exp(\beta_0), & t = 1, \ 2, \ \dots, \ v - 1 \\
Exp(\beta_1), & t = v, \ v + 1, \ \dots
\end{cases}
\tag{6}
$$

where $\beta_0$ and $\beta_1$ are known parameters; $\beta_1 > \beta_0$. As mentioned previously, when considering the change point in Eq (6), the ARL defined with $E_v(.)$ can be characterized as follows:

$$
\text{ARL} =
\begin{cases}
\text{ARL}_0 = E_\infty(\tau_b), & v = \infty \ (no \ change) \\
\text{ARL}_1 = E_1(\tau_b), & v = 1 \ (change)
\end{cases}
,
$$

where $E_v()$ denotes the expectation under distribution $F(x, \beta)$ for a given change-point time (*v*). $v = \infty$ indicates no change in the statistical process and is referred to as the in-control ARL (ARL$_0$). Herein, $v = 1$ marks the first time point that a change takes place from $\beta_0$ to $\beta$ in the statistical process, which is called out-of-control ARL (ARL$_1$).

**Definitions and theories for evaluating the ARL.**   Definitions, theories, and concepts necessary to prove the existence and uniqueness of the IEs in the proof based on the functional analysis are covered in this section.

**Definition 2**. Let (*M*, *d*) be a metric space and let $\Upsilon: M \to M$ be a mapping. A point $u \in M$ is called a fixed point of $\Upsilon$ if $\Upsilon(u) = u$.

**Remark**: Fixed point theorems guarantee the existence of a fixed point under appropriate conditions on the map $\Upsilon$ and the set *M*.

**Definition 3**. A normed vector space *M* is a Banach space if the metric space (*M*, *d*) is complete, where $d(u, v) = \|u - v\|$, for all $u, v \in M$.

Consider the Banach space is *n*-dimensional Euclidean space $\mathbb{R}^n$, where the norm $|.|$ is given by the Euclidean distance. The $\mathbf{C}(M)$ is space of continuous functions where $M$ is a compact interval and the norm $\|.\|$ is given by $\|f\|_\infty = \sup_{u \in M} |f(u)|$ for $f \in \mathbf{C}(M)$.

**Definition 4**. Let $(M, d)$ be a metric space and let $\Upsilon: M \to M$ be a map, and $\Upsilon$ is called contraction if there exists a fixed constant $\rho \in [0, 1)$ such that

$$d(\Upsilon(u), \Upsilon(v)) \leq \rho d(u, v), \text{ for all } u, v \in M.$$

**Theorem 1 (Banach's fixed-point theorem or the contraction theorem)** [27]. Let $(M, d)$ be a complete metric space, then each contraction map $\Upsilon: M \to M$ has a unique fixed point.

## The existence and uniqueness of the analytical IE for the ARL on a CUSUM control chart

The analytical IE for the ARL of a CUSUM control chart running a long-memory SARFIMAX process corresponds to a Fredholm integral equation of the second type. In this section, Banach's fixed-point theorem is applied to prove the existence and uniqueness of the analytical IE solution and demonstrate that it has the same accuracy as the numerical IE method.

Throughout, this paper, let $\ell(\varphi)$ be the ARL conditioned on the initial value for a CUSUM control chart running a long-memory SARFIMAX process with underlying exponential white noise [17]. The initial value for monitoring CUSUM statistic $C_0 = \varphi$ is determined at $\varphi \in [0, b]$. Let $\mathbb{P}_c$ and $\mathbb{E}_c$ be the probability measure and induced expectation corresponding to $\varphi$. Moreover, ARL is defined as a function of $\ell(\varphi) = \mathbb{E}_c(\tau_b) < \infty$, where $\tau_b = \inf\{t > 0; C_t > b\}$ satisfies the solution for the analytical IE as follows:

$$\ell(\varphi) = 1 + \mathbb{P}_c\{C_1 = 0\}\ell(0) + \mathbb{E}_c[I\{0 < C_1 < b\}\ell(C_1)], \tag{7}$$

where indicator function is $I\{0 < C_1 < b\} = \begin{cases} 1; & 0 < C_1 < b \\ 0; & \text{Otherwise.} \end{cases}$

The IE for the ARL derived from the Fredholm integral equation of the second kind can be written as follows:

$$\ell(\varphi) = 1 + \ell(0)F(\eta - \varphi - Y_t) + \int_0^b \ell(g)f(g + \eta - \varphi - Y_t)dg, \tag{8}$$

where $f(.)$ is the probability density function (pdf) of an exponential distribution [17].

Hence, deriving the analytical IE via Eq (8) becomes

$$
\begin{aligned}
\ell(\varphi) \;=\; & 1 + (1 - \exp\left\{ \begin{array}{l} -\beta\left(\eta - \varphi - \mu - \xi_t + \sum_{j=1}^{Q}\Theta_j\xi_{t-js} - \sum_{l=1}^{r}\omega_l X_{lt} - \left(DY_{t-s} - \dfrac{D(D-1)}{2!}Y_{t-2s} + \dfrac{D(D-1)(D-2)}{3!}Y_{t-3s} - \cdots\right)\right) \\[2ex] -\sum_{i=1}^{P}\Phi_i\left(Y_{t-is} - DY_{t-(i+1)s} + \dfrac{D(D-1)}{2!}Y_{t-(i+2)s} - \dfrac{D(D-1)(D-2)}{3!}Y_{t-(i+3)s} + \cdots\right)\right) \end{array} \right\})\ell(0) \\[4ex]
& + \beta(\exp\left\{ \begin{array}{l} \beta\left(\varphi - \eta + \mu + \xi_t - \sum_{j=1}^{Q}\Theta_j\xi_{t-js} + \sum_{l=1}^{r}\omega_l X_{lt} + \left(DY_{t-s} - \dfrac{D(D-1)}{2!}Y_{t-2s} + \dfrac{D(D-1)(D-2)}{3!}Y_{t-3s} - \cdots\right)\right) \\[2ex] +\sum_{i=1}^{P}\Phi_i\left(Y_{t-is} - DY_{t-(i+1)s} + \dfrac{D(D-1)}{2!}Y_{t-(i+2)s} - \dfrac{D(D-1)(D-2)}{3!}Y_{t-(i+3)s} + \cdots\right)\right) \end{array} \right\})\int_0^b \ell(g)\exp\{-\beta g\}dg, \quad \varphi \in [0, b],
\end{aligned}
\tag{9}
$$

where $\ell(.)$ is an arbitrary function on $[0, b]$. The right-hand side of Eq (9) becomes a

continuous function, and so theoretically, it is clear that the ARL function $\ell(.)$ is a continuous solution on $[0, b]$. for the IE in Eq (9). After that, definitions and Banach's fixed-point theorem are applied to guarantee the existence and uniqueness of the solution for functional $\ell(\varphi)$.

**Theorem 2**. The existence and uniqueness of the solution for the analytical IE are derived for a long-memory SARFIMAX$(P, D, Q, r)_s$ process with underlying exponential white noise.

**Proof: (Existence)**. As shown in Appendix A in S1 File,

$$\Upsilon(\ell) = \lim_{n \to \infty} \Upsilon(\ell_n) = \lim_{n \to \infty} \ell_{n+1} = \ell.$$

In this situation we have

$$\Upsilon(\ell(\varphi)) = 1 + (1 - \exp\left\{ \begin{aligned} &-\beta\left(\eta - \varphi - \mu - \xi_t + \sum_{j=1}^{Q}\Theta_j\xi_{t-js} - \sum_{l=1}^{r}\omega_l X_{lt} - \left(DY_{t-s} - \frac{D(D-1)}{2!}Y_{t-2s} + \frac{D(D-1)(D-2)}{3!}Y_{t-3s} - \ldots\right)\right) \\ &-\sum_{i=1}^{P}\Phi_i\left(Y_{t-is} - DY_{t-(i+1)s} + \frac{D(D-1)}{2!}Y_{t-(i+2)s} - \frac{D(D-1)(D-2)}{3!}Y_{t-(i+3)s} + \ldots\right) \end{aligned} \right\})\ell(0)$$
$$+ \beta(\exp\left\{ \begin{aligned} &\beta(\varphi - \eta + \mu + \xi_t - \sum_{j=1}^{Q}\Theta_j\xi_{t-js} + \sum_{l=1}^{r}\omega_l X_{lt} + \left(DY_{t-s} - \frac{D(D-1)}{2!}Y_{t-2s} + \frac{D(D-1)(D-2)}{3!}Y_{t-3s} - \ldots\right) \\ &+\sum_{i=1}^{P}\Phi_i\left(Y_{t-is} - DY_{t-(i+1)s} + \frac{D(D-1)}{2!}Y_{t-(i+2)s} - \frac{D(D-1)(D-2)}{3!}Y_{t-(i+3)s} + \ldots\right) \end{aligned} \right\})\int_0^b \ell(g)\exp\{-\beta g\}dg,$$

where $\ell(\varphi)$ is a fixed point of $\Upsilon$. Hence, the IE in Eq (9) can be rewritten as $\Upsilon(\ell(\varphi)) = \ell(\varphi)$, which is Banach's fixed-point theorem. If operator $\Upsilon$ is a contraction, then fixed point $\Upsilon(\ell(\varphi)) = \ell(\varphi)$.

**Proof: (Uniqueness)**. Let $\Upsilon$ be a contraction mapping on a complete metric space. By considering Banach's fixed-point theorem, it can be shown that $\Upsilon: \mathbf{C}([0, b]) \to \mathbf{C}([0, b])$ is a contraction mapping on the complete metric space $(\mathbf{C}([0, b]), \|.\|_\infty)$ where metric $\mathbf{C}([0, b])$ is the space of all continuous functions $\ell(\varphi)$ on interval; $[0, b]$ endowed with supremum norm $\|.\|_\infty = \sup_{\varphi \in [0, b]} |\ell(\varphi)|$, for every function $\ell(.) \in \mathbf{C}([0, b])$. It has a unique solution, as shown in Appendix B in S1 File.

Thus, as proved and guaranteed by applying Banach's fixed-point theorem, the analytical IE for a long-memory SARFIMAX$(P, D, Q, r)_s$ process with underlying exponential white noise on the CUSUM control chart exists and is unique for solutions of functional $\ell(\varphi)$.

## Analytical and approximate ARLs for a long-memory SARFIMAX process with underlying exponential white noise on a CUSUM control chart

In this section, we apply the proposed analytical and numerical IEs corresponding to a Fredholm integral equation of the second type representing analytical and approximate ARLs for monitoring changes in the process mean of a long-memory SARFIMAX process with underlying exponential white noise on a CUSUM control chart.

## The proposed analytical IE for a SARFIMAX(*P*, *D*, *Q*, *r*)$_s$ process with underlying white noise on a CUSUM control chart

**Theorem 3**. Exact formula $\ell(\varphi)$ for monitoring changes in the process mean of a long-memory SARFIMAX process with underlying exponential white noise on a CUSUM control charts can

be expressed as

$$
\ell(\varphi) = \exp\{\beta b\}((1 + \exp\left\{ \begin{array}{l} \beta\left(\eta - \mu - \xi_t + \sum_{j=1}^{Q}\Theta_j\xi_{t-js} - \sum_{l=1}^{r}\omega_l X_{lt} - \left(DY_{t-s} - \dfrac{D(D-1)}{2!}Y_{t-2s} + \dfrac{D(D-1)(D-2)}{3!}Y_{t-3s} - \ldots\right)\right) \\[2ex] -\sum_{i=1}^{P}\Phi_i\left(Y_{t-is} - DY_{t-(i+1)s} + \dfrac{D(D-1)}{2!}Y_{t-(i+2)s} - \dfrac{D(D-1)(D-2)}{3!}Y_{t-(i+3)s} + \ldots\right)\right) \end{array} \right\} - \beta b)
$$

$$
- \exp\{\beta\varphi\}, \quad \text{for } \varphi \geq 0.
$$

**Proof**: Let $\ell(\varphi)$ be the ARL of the IE corresponding to the long-memory SARFIMAX(*P*, *D*, *Q*, *r*)$_s$ process on a CUSUM control chart. Thus, $c = \int_0^b \ell(g)\exp\{-\beta g\}dg$ can be substituted as follows:

$$
\ell(\varphi) = 1 + (1 - \exp\left\{ \begin{array}{l} -\beta\left(\eta - \varphi - \mu - \xi_t + \sum_{j=1}^{Q}\Theta_j\xi_{t-js} - \sum_{l=1}^{r}\omega_l X_{lt} - \left(DY_{t-s} - \dfrac{D(D-1)}{2!}Y_{t-2s} + \dfrac{D(D-1)(D-2)}{3!}Y_{t-3s} - \ldots\right)\right) \\[2ex] -\sum_{i=1}^{P}\Phi_i\left(Y_{t-is} - DY_{t-(i+1)s} + \dfrac{D(D-1)}{2!}Y_{t-(i+2)s} - \dfrac{D(D-1)(D-2)}{3!}Y_{t-(i+3)s} + \ldots\right)\right) \end{array} \right\})\ell(0)
$$

$$
+ c\beta(\exp\left\{ \begin{array}{l} \beta\left(\varphi - \eta + \mu + \xi_t - \sum_{j=1}^{Q}\Theta_j\xi_{t-js} + \sum_{l=1}^{r}\omega_l X_{lt} + \left(DY_{t-s} - \dfrac{D(D-1)}{2!}Y_{t-2s} + \dfrac{D(D-1)(D-2)}{3!}Y_{t-3s} - \ldots\right)\right) \\[2ex] +\sum_{i=1}^{P}\Phi_i\left(Y_{t-is} - DY_{t-(i+1)s} + \dfrac{D(D-1)}{2!}Y_{t-(i+2)s} - \dfrac{D(D-1)(D-2)}{3!}Y_{t-(i+3)s} + \ldots\right)\right) \end{array} \right\}). \tag{10}
$$

By replacing $\varphi = 0$ in Eq (10), we obtain

$$
\ell(\varphi) = 1 + (1 - \exp\left\{ \begin{array}{l} -\beta\left(\eta - \mu - \xi_t + \sum_{j=1}^{Q}\Theta_j\xi_{t-js} - \sum_{l=1}^{r}\omega_l X_{lt} - \left(DY_{t-s} - \dfrac{D(D-1)}{2!}Y_{t-2s} + \dfrac{D(D-1)(D-2)}{3!}Y_{t-3s} - \ldots\right)\right) \\[2ex] -\sum_{i=1}^{P}\Phi_i\left(Y_{t-is} - DY_{t-(i+1)s} + \dfrac{D(D-1)}{2!}Y_{t-(i+2)s} - \dfrac{D(D-1)(D-2)}{3!}Y_{t-(i+3)s} + \ldots\right)\right) \end{array} \right\})\ell(0)
$$

$$
+ c\beta(\exp\left\{ \begin{array}{l} \beta\left(\mu - \eta + \xi_t - \sum_{j=1}^{Q}\Theta_j\xi_{t-js} + \sum_{l=1}^{r}\omega_l X_{lt} + \left(DY_{t-s} - \dfrac{D(D-1)}{2!}Y_{t-2s} + \dfrac{D(D-1)(D-2)}{3!}Y_{t-3s} - \ldots\right)\right) \\[2ex] +\sum_{i=1}^{P}\Phi_i\left(Y_{t-is} - DY_{t-(i+1)s} + \dfrac{D(D-1)}{2!}Y_{t-(i+2)s} - \dfrac{D(D-1)(D-2)}{3!}Y_{t-(i+3)s} + \ldots\right)\right) \end{array} \right\}).
$$

$$
\therefore \ell(0) = c\beta + \exp\left\{ \begin{array}{l} \beta\left(\eta - \mu - \xi_t + \sum_{j=1}^{Q}\Theta_j\xi_{t-js} - \sum_{l=1}^{r}\omega_l X_{lt} - \left(DY_{t-s} - \dfrac{D(D-1)}{2!}Y_{t-2s} + \dfrac{D(D-1)(D-2)}{3!}Y_{t-3s} - \ldots\right)\right) \\[2ex] -\sum_{i=1}^{P}\Phi_i\left(Y_{t-is} - DY_{t-(i+1)s} + \dfrac{D(D-1)}{2!}Y_{t-(i+2)s} - \dfrac{D(D-1)(D-2)}{3!}Y_{t-(i+3)s} + \ldots\right)\right) \end{array} \right\} \tag{11}
$$

Subsequently, by substituting $\ell(0)$ into Eq (10), it follows that

$$\ell(\varphi) = 1 + c\beta + \exp\left\{ \begin{array}{c} \beta\left(\eta - \mu - \xi_t + \sum_{j=1}^{Q}\Theta_j\xi_{t-js} - \sum_{l=1}^{r}\omega_l X_{lt} - \left(DY_{t-s} - \frac{D(D-1)}{2!}Y_{t-2s} + \frac{D(D-1)(D-2)}{3!}Y_{t-3s} - \cdots\right)\right. \\ \left. -\sum_{i=1}^{P}\Phi_i\left(Y_{t-is} - DY_{t-(i+1)s} + \frac{D(D-1)}{2!}Y_{t-(i+2)s} - \frac{D(D-1)(D-2)}{3!}Y_{t-(i+3)s} + \cdots\right)\right) \end{array} \right\} - \exp\{\beta\varphi\}. \quad (12)$$

Constant $c = \int_0^b \ell(g)\exp\{-\beta g\}dg$, can be derived as

$$c = (1 + c\beta + \exp\left\{ \begin{array}{c} \beta\left(\eta - \mu - \xi_t + \sum_{j=1}^{Q}\Theta_j\xi_{t-js} - \sum_{l=1}^{r}\omega_l X_{lt} - \left(DY_{t-s} - \frac{D(D-1)}{2!}Y_{t-2s} + \frac{D(D-1)(D-2)}{3!}Y_{t-3s} - \cdots\right)\right. \\ \left. -\sum_{i=1}^{P}\Phi_i\left(Y_{t-is} - DY_{t-(i+1)s} + \frac{D(D-1)}{2!}Y_{t-(i+2)s} - \frac{D(D-1)(D-2)}{3!}Y_{t-(i+3)s} + \cdots\right)\right) \end{array} \right\})$$

$$\int_0^b \exp\{-\beta g\}dg - \int_0^b \exp\{\beta g - \beta g\}dg$$

After rearranging, we obtain

$$\therefore c = \beta^{-1}\exp\{\beta b\}(1 + \exp\left\{ \begin{array}{c} \beta\left(\eta - \mu - \xi_t + \sum_{j=1}^{Q}\Theta_j\xi_{t-js} - \sum_{l=1}^{r}\omega_l X_{lt} - \left(DY_{t-s} - \frac{D(D-1)}{2!}Y_{t-2s} + \frac{D(D-1)(D-2)}{3!}Y_{t-3s} - \cdots\right)\right. \\ \left. -\sum_{i=1}^{P}\Phi_i\left(Y_{t-is} - DY_{t-(i+1)s} + \frac{D(D-1)}{2!}Y_{t-(i+2)s} - \frac{D(D-1)(D-2)}{3!}Y_{t-(i+3)s} + \cdots\right)\right) \end{array} \right\}) \quad (13)$$

$$\times(1 - \exp\{-\beta b\}) - b\exp\{\beta b\}.$$

Finally, by substituting constant $c$ from Eq (13) into Eq (12), we obtain

$$\ell(\varphi) = 1 + (\exp\{\beta b\}(1 - \exp\{-\beta b\})(1 + \exp\left\{ \begin{array}{c} \beta\left(\eta - \mu - \xi_t + \sum_{j=1}^{Q}\Theta_j\xi_{t-js} - \sum_{l=1}^{r}\omega_l X_{lt} - \left(DY_{t-S} - \frac{D(D-1)}{2!}Y_{t-2s} + \frac{D(D-1)(D-2)}{3!}Y_{t-3s} - \cdots\right)\right. \\ \left. -\sum_{i=1}^{P}\Phi_i\left(Y_{t-is} - DY_{t-(i+1)s} + \frac{D(D-1)}{2!}Y_{t-(i+2)s} - \frac{D(D-1)(D-2)}{3!}Y_{t-(i+3)s} + \cdots\right)\right) \end{array} \right\})$$

$$-\beta b\exp\{\beta b\})$$

$$\ell(\varphi) = \exp\{\beta b\}(1 + \exp\left\{ \begin{array}{c} \beta\left(\eta - \mu - \xi_t + \sum_{j=1}^{Q}\Theta_j\xi_{t-js} - \sum_{l=1}^{r}\omega_l X_{lt} - \left(DY_{t-s} - \frac{D(D-1)}{2!}Y_{t-2s} + \frac{D(D-1)(D-2)}{3!}Y_{t-3s} - \cdots\right)\right. \\ \left. -\sum_{i=1}^{P}\Phi_i\left(Y_{t-is} - DY_{t-(i+1)s} + \frac{D(D-1)}{2!}Y_{t-(i+2)s} - \frac{D(D-1)(D-2)}{3!}Y_{t-(i+3)s} + \cdots\right)\right) \end{array} \right\} - \beta b) \quad (14)$$

$$-\exp\{\beta\varphi\}; \quad \varphi \geq 0.$$

The proof is complete.

As shown in the previous equation, the in-control ARL (ARL$_0$) is assigned exponential parameter ($\beta = \beta_0$), as follows:

$$
\mathrm{ARL}_0 = \exp\{\beta_0 b\}(1 + \exp\left\{ \begin{aligned} &\beta_0\left(\eta - \mu - \xi_t + \sum_{j=1}^{Q}\Theta_j\xi_{t-js} - \sum_{l=1}^{r}\omega_l X_{lt} - \left(DY_{t-s} - \frac{D(D-1)}{2!}Y_{t-2s} + \frac{D(D-1)(D-2)}{3!}Y_{t-3s} - \cdots\right)\right.\\ &\left. -\sum_{i=1}^{P}\Phi_i\left(Y_{t-is} - DY_{t-(i+1)s} + \frac{D(D-1)}{2!}Y_{t-(i+2)s} - \frac{D(D-1)(D-2)}{3!}Y_{t-(i+3)s} + \cdots\right)\right) \end{aligned} \right\} - \beta_0 b) - \exp\{\beta_0\varphi\}. \quad (15)
$$

Similarly, the out-of-control ARL (ARL$_1$) is assigned exponential parameter ($\beta = \beta_1$), where $\beta_1 = \beta_0(1 + \delta)$, and the shift in the mean is determined by $\delta$ and $\beta_0 = 1$ as follows:

$$
\mathrm{ARL}_1 = \exp\{\beta_1 b\}(1 + \exp\left\{ \begin{aligned} &\beta_1\left(\eta - \mu - \xi_t + \sum_{j=1}^{Q}\Theta_j\xi_{t-js} - \sum_{l=1}^{r}\omega_l X_{lt} - \left(DY_{t-s} - \frac{D(D-1)}{2!}Y_{t-2s} + \frac{D(D-1)(D-2)}{3!}Y_{t-3s} - \cdots\right)\right.\\ &\left. -\sum_{i=1}^{P}\Phi_i\left(Y_{t-is} - DY_{t-(i+1)s} + \frac{D(D-1)}{2!}Y_{t-(i+2)s} - \frac{D(D-1)(D-2)}{3!}Y_{t-(i+3)s} + \cdots\right)\right) \end{aligned} \right\} - \beta_1 b) - \exp\{\beta_1\varphi\}. \quad (16)
$$

These circumstances show that the computational scheme can be completed in one stage.

## The numerical ARL for a SARFIMAX(*P*, *D*, *Q*, *r*)$_s$ process with underlying white noise on a CUSUM control chart

An approximate ARL using the numerical IE method via Fredholm's integral equation of the second kind was previously formulated for a CUSUM control chart by Peerajit et al. [22]. For this method, an approximate ARL was developed by using the Gauss-Legendre quadrature rules technique.

**Definition 5**. The quadrature rules usually applied to integral $\int_0^b f(g)dg$ can be approximated by the sum of the areas of a rectangle is as follows:

$$
\int_0^b W(g)f(g)dg \approx \sum_{j=1}^{m} w_j f(v_j), \text{ with } v_j = \frac{b}{m}\left(j - \frac{1}{2}\right), \; j = 1, 2, \ldots, m,
$$

where the value of integral $f$ is chosen by applying base $b/m$ with heights at the midpoints of the intervals of length $b/m$ beginning at zero and $W(g)$ is a weight function. Division points $v_1 \leq \ldots \leq v_m$ fall within interval $[0, b]$ and $w_j = b/m \geq 0$ set of constant weights $w_j, j = 1, 2, \ldots, m$.

**Theorem 4**. The numerical IE method ($\vartheta(\varphi)$) for monitoring changes in the process mean of a long-memory SARFIMAX process with underlying exponential white noise ($Y_t$) on a CUSUM control chart can be expressed as

$$
\vartheta(\varphi) = 1 + \vartheta(v_1)F(\eta - \varphi - Y_t) + \sum_{j=1}^{m} w_j\vartheta(v_j)f(v_j + \eta - \varphi - Y_t),
$$

with $w_j = \frac{b}{m}$, and $v_j = \frac{b}{m}\left(j - \frac{1}{2}\right)$ ; $j = 1, 2, \ldots, m$.

***Proof***: By applying $\vartheta(\varphi)$ as the approximate ARL from the IE solution in Eq (9) using the Gauss-Legendre quadrature rule technique and replacing $\varphi$ with $v_j$, we obtain

$$\vartheta(v_i) = 1 + \vartheta(v_1)F(\eta - v_i - Y_t) + \sum_{j=1}^{m} w_j\vartheta(v_j)f(v_j + \eta - v_i - Y_t), \tag{17}$$

With the system of $m$ linear equations with $m$ unknowns $\vartheta(v_1), \vartheta(v_2), \ldots, \vartheta(v_m)$, we can rearrange Eq (17) as follows:

$$\vartheta(v_1) = 1 + \vartheta(v_1)[F(\eta - v_1 - Y_t) + w_1 f(\eta - Y_t)] + \sum_{j=2}^{m} w_j\vartheta(v_j)f(v_j + \eta - v_1 - Y_t)$$

$$\vartheta(v_2) = 1 + \vartheta(v_1)[F(\eta - v_2 - Y_t) + w_1 f(v_1 + \eta - v_2 - Y_t)] + \sum_{j=2}^{m} w_j\vartheta(v_j)f(v_j + \eta - v_2 - Y_t)$$

$$\vdots$$

$$\vartheta(v_m) = 1 + \vartheta(v_1)[F(\eta - v_m - Y_t) + w_1 f(v_1 + \eta - v_m - Y_t)] + \sum_{j=2}^{m} w_j\vartheta(v_j)f(v_j + \eta - v_m - Y_t)$$

This is equivalent to matrix form $\mathbf{J}_{m\times1} = \mathbf{1}_{m\times1} + \mathbf{R}_{m\times m}\mathbf{J}_{m\times1}$ If the inverse $(\mathbf{I}_m - \mathbf{R}_{m\times m})^{-1}$ exists, then the unique solution is

$$\mathbf{J}_{m\times1} = (\mathbf{I}_m - \mathbf{R}_{m\times m})^{-1}\mathbf{1}_{m\times1},$$

where $\mathbf{J}_{m\times1} = [\vartheta(v_1), \vartheta(v_2), \ldots, \vartheta(v_m)]'$, $\mathbf{I}_m = diag(1, 1, \ldots, 1)$ is the unit matrix order $m$, $\mathbf{1}_{m\times1} = [1, 1, \ldots, 1]'$ is a column vector of $\vartheta(v_j)$, and $\mathbf{R}_{m\times m}$ is a matrix of $(m, m)^{th}$ elements:

$$\mathbf{R}_{m\times m} = \begin{bmatrix} r_{11} & r_{12} & \cdots & r_{1m} \\ r_{21} & r_{22} & \cdots & r_{2m} \\ \vdots & \vdots & \ddots & \vdots \\ r_{m1} & r_{m2} & \cdots & r_{mm} \end{bmatrix}$$

where $r_{ij} = F(\eta - v_i - Y_t) + w_j f(v_j + \eta - v_i - Y_t)$ ; $i, j = 1, 2, \ldots, m$.

Therefore, the approximation for the integral is obtained in the summation where $v_j$ is replaced by $\varphi$ in $\vartheta(v_j)$, which completes the proof.

## Performance evaluation and comparison

A popular benchmark to measure the performance of a control chart is the ARL for detecting shifts in the process mean as a measure of its sensitivity and practicability. Thus, the performances of the analytical and numerical IE methods for determining the ARL for monitoring shifts in the process mean for a long-memory SARFIMAX(*P, D, Q, r*)$_s$ process on a CUSUM control chart were compared. We determined the number of division points $m = 800$ for the approximated ARL by using the numerical IE method. The in-control situation comprises exponential white noise with mean $\beta_0 = 1$, while the out-of-control process is signaled by a shift in the process mean from $\beta_0$ to $\beta_1$, where $\beta_1 = \beta_0(1 + \delta)$. Thus, $\delta = 0.01, 0.03, 0.05, 0.10, 0.30, 0.50, 0.70, 1.00, 1.50,$ and $2.00$. $\delta = 0$, means that the process is in-control whereas $\delta > 0$, means that the process is out-of-control, denoted as ARL$_0$ and ARL$_1$, respectively. Accordingly, ARL$_0$ was fixed at 370 or 500. For the CUSUM statistic, the value of $\eta$ was determined as

**Table 1. Calculated $b$ for corresponding reference parameter $\eta$ values for a long-memory SARFIMAX($P$, $D$, $Q$, $r$)$_s$ process on a CUSUM control chart for ARL$_0$ = 370, 500.**

| ARL$_0$ | Coefficients of models | | | | Long-memory Models | $\eta$ | | |
|---|---|---|---|---|---|---|---|---|
| | $\Phi_1$ | $\Phi_2$ | $\Phi_1$ | $\omega_1$ | | 3.0 | 3.5 | 4.0 |
| 370 | 0.1 | 0.2 | 0.1 | 0.5 | [1] SARFIMAX(2, 0.15, 1, 1)$_{12}$ | 4.300562 | 3.498061 | 2.884593 |
| | 0.1 | 0.2 | 0.1 | 0.5 | [2] SARFIMAX(2, 0.30, 1, 1)$_{12}$ | 4.653230 | 3.704270 | 3.056922 |
| | 0.1 | | 0.1 | 0.5 | [3] SARFIMAX(1, 0.45, 1, 1)$_{12}$ | 5.091801 | 3.888005 | 3.202990 |
| | -0.1 | 0.2 | 0.1 | 0.5 | [−1] SARFIMAX(2, 0.15, 1, 1)$_{12}$ | 4.021805 | 3.304131 | 2.715860 |
| | -0.1 | 0.2 | 0.1 | 0.5 | [−2] SARFIMAX(2, 0.30, 1, 1)$_{12}$ | 4.387763 | 3.553436 | 2.931650 |
| | -0.1 | | 0.1 | 0.5 | [−3] SARFIMAX(1, 0.45, 1, 1)$_{12}$ | 4.802330 | 3.776275 | 3.115083 |
| 500 | 0.1 | 0.2 | 0.1 | 0.5 | [1] SARFIMAX(2, 0.15, 1, 1)$_{12}$ | 4.677214 | 3.826880 | 3.199020 |
| | 0.1 | 0.2 | 0.1 | 0.5 | [2] SARFIMAX(2, 0.30, 1, 1)$_{12}$ | 5.080630 | 4.040850 | 3.374410 |
| | 0.1 | | 0.1 | 0.5 | [3] SARFIMAX(1, 0.45, 1, 1)$_{12}$ | 5.698833 | 4.233519 | 3.523590 |
| | -0.1 | 0.2 | 0.1 | 0.5 | [−1] SARFIMAX(2, 0.15, 1, 1)$_{12}$ | 4.375410 | 3.627221 | 3.027800 |
| | -0.1 | 0.2 | 0.1 | 0.5 | [−2] SARFIMAX(2, 0.30, 1, 1)$_{12}$ | 4.774041 | 3.884130 | 3.246850 |
| | -0.1 | | 0.1 | 0.5 | [−3] SARFIMAX(1, 0.45, 1, 1)$_{12}$ | 5.266000 | 4.116095 | 3.433746 |

3.0, 3.5, or 4.0. Thus, the CUSUM control limit ($b$) was calculated by using Eq (15) to achieve the specified ARL, the results for which are reported in Table 1. The coefficient parameters for long-memory SARFIMAX(2, 0.15, 1, 1)$_{12}$, SARFIMAX(2, 0.30, 1, 1)$_{12}$ and SARFIMAX(1, 0.45, 1, 1)$_{12}$ processes were set as $\Phi_1 = \pm 0.1$, $\Phi_2 = 0.2$, $\Phi_1 = 0.1$, and $\omega_1 = 0.5$. For a fixed ARL$_0$, the method achieving the smallest ARL$_1$ has delivered the best performance [1]. The results for the sensitivity, accuracy (see Definition 6), and computational times of the two methods are provided in Tables 2–6 and Fig 1.

**Definition 6**. The percentage accuracy (% Accuracy) indicates the relative performances of the analytical and numerical IE methods defined as follows:

$$\%\text{Accuracy} = 100 - \left| \frac{\ell(\varphi) - \vartheta(\varphi)}{\ell(\varphi)} \right| \times 100\%,$$

where $\ell(\varphi)$ and $\vartheta(\varphi)$ are the ARL values obtained using the analytical and numerical IE methods, respectively.

The following algorithm was developed by using the Mathematica program to calculate the values of the ARLs for a long-memory SARFIMAX($P$, $D$, $Q$, $r$)$_s$ process on a CUSUM control chart:

Step 1: Determine the values of coefficients $\Phi_1$, $\Phi_2$, $\Theta_1$, $\omega_1$, and the initial values of $\xi_{t-s}$, $\xi_{t-2s}$, . . ., $\xi_{t-Qs}$, $X_{lt}$, $Y_{t-s}$, $Y_{t-2s}$, $Y_{t-3s}$, . . ., $Y_{t-Ps}$, $Y_{t-(P+1)s}$.

Step 2: Specify $\beta_0$ for the in-control exponential white noise ($\xi_t \sim Exp(\beta)$).

Step 3: Determine the value of known parameter ($\eta$) and the initial value of CUSUM statistic $\varphi$.

Step 4: Calculate the upper control limit ($b$) by using Eq (15) for the specified ARL$_0$ and $\eta$.

Step 5: Compute ARL$_1$ for shifts in the process mean ($\beta_1$) where $b$ is attained from Step 4.

The above algorithm was executed to obtain ARLs for out-of-control situations as measurements of the performance of the proposed analytical and numerical IE methods.

The results for computed parameter $b$ and reference parameter $\eta$ for ARL$_0$ = 370 or 500 using the proposed ARL schemes are reported in Table 1. $b$ is inversely proportional to $\eta$ for

**Table 2. Comparison of the $ARL_1$ values obtained using the analytical and numerical ARL methods for long-memory SARFIMAX$(P, D, Q, r)_s$ processes with exponential white noise on a CUSUM control chart for $ARL_0 = 370$ with $\Phi_1 = 0.1$, $\Phi_2 = 0.2$, $\Theta_1 = 0.1$, and $\omega_1 = 0.5$.**

| $\delta$ | $\eta$ | 3.0 | | | 3.5 | | | 4.0 | | |
|---|---|---|---|---|---|---|---|---|---|---|
| | $b$ | 4.300562 | 4.65323 | 5.091801 | 3.498061 | 3.70427 | 3.888005 | 2.884593 | 3.056922 | 3.20299 |
| | Model | [1] | [2] | [3] | [1] | [2] | [3] | [1] | [2] | [3] |
| 0.01 | $\ell(\varphi)$ | 345.218 | 343.883 | 341.545 | 347.171 | 346.781 | 346.375 | 348.003 | 347.812 | 347.626 |
| | (Sec.) | (0.001) | (0.001) | (0.001) | (0.001) | (0.001) | (0.001) | (0.001) | (0.001) | (0.001) |
| | $\vartheta(\varphi)$ | 344.460 | 343.135 | 340.870 | 346.481 | 346.065 | 345.639 | 347.410 | 347.190 | 346.979 |
| | (Min.) | (32.22) | (32.11) | (31.81) | (31.99) | (32.62) | (31.65) | (31.89) | (32.76) | (31.70) |
| | %Accuracy | 99.78 | 99.78 | 99.80 | 99.80 | 99.79 | 99.79 | 99.83 | 99.82 | 99.81 |
| 0.03 | $\ell(\varphi)$ | 301.755 | 298.353 | 292.454 | 306.773 | 305.765 | 304.718 | 308.935 | 308.438 | 307.954 |
| | (Sec.) | (0.001) | (0.001) | (0.001) | (0.001) | (0.001) | (0.001) | (0.001) | (0.001) | (0.001) |
| | $\vartheta(\varphi)$ | 301.116 | 297.734 | 291.919 | 306.179 | 305.151 | 304.090 | 308.421 | 307.899 | 307.395 |
| | (Min.) | (31.99) | (31.93) | (31.77) | (32.09) | (32.64) | (31.64) | (32.11) | (32.82) | (31.76) |
| | %Accuracy | 99.79 | 99.79 | 99.82 | 99.81 | 99.80 | 99.79 | 99.83 | 99.83 | 99.82 |
| 0.05 | $\ell(\varphi)$ | 265.144 | 260.308 | 251.990 | 272.343 | 270.888 | 269.383 | 275.478 | 274.756 | 274.053 |
| | (Sec.) | (0.001) | (0.001) | (0.001) | (0.001) | (0.001) | (0.001) | (0.001) | (0.001) | (0.001) |
| | $\vartheta(\varphi)$ | 264.602 | 259.793 | 251.564 | 271.830 | 270.359 | 268.844 | 275.031 | 274.287 | 273.567 |
| | (Min.) | (31.72) | (32.44) | (31.76) | (31.75) | (32.31) | (31.46) | (31.73) | (32.02) | (31.66) |
| | %Accuracy | 99.80 | 99.80 | 99.83 | 99.81 | 99.80 | 99.80 | 99.84 | 99.83 | 99.82 |
| 0.10 | $\ell(\varphi)$ | 195.987 | 189.360 | 178.168 | 206.079 | 204.013 | 201.891 | 210.588 | 209.541 | 208.526 |
| | (Sec.) | (0.001) | (0.001) | (0.001) | (0.001) | (0.001) | (0.001) | (0.001) | (0.001) | (0.001) |
| | $\vartheta(\varphi)$ | 195.621 | 189.027 | 177.924 | 205.714 | 203.640 | 201.515 | 210.265 | 209.203 | 208.178 |
| | (Min.) | (31.64) | (31.91) | (31.65) | (31.55) | (33.22) | (31.56) | (31.49) | (31.86) | (31.48) |
| | %Accuracy | 99.81 | 99.82 | 99.86 | 99.82 | 99.82 | 99.81 | 99.85 | 99.84 | 99.83 |
| 0.30 | $\ell(\varphi)$ | 74.675 | 69.144 | 60.371 | 83.787 | 81.838 | 79.886 | 88.238 | 87.176 | 86.165 |
| | (Sec.) | (0.001) | (0.001) | (0.001) | (0.001) | (0.001) | (0.001) | (0.001) | (0.001) | (0.001) |
| | $\vartheta(\varphi)$ | 74.576 | 69.068 | 60.343 | 83.671 | 81.722 | 79.772 | 88.129 | 87.063 | 86.050 |
| | (Min.) | (31.70) | (31.78) | (31.45) | (31.59) | (31.99) | (31.74) | (31.71) | (31.90) | (31.81) |
| | %Accuracy | 99.87 | 99.89 | 99.95 | 99.86 | 99.86 | 99.86 | 99.88 | 99.87 | 99.87 |
| 0.50 | $\ell(\varphi)$ | 37.626 | 34.278 | 29.236 | 43.515 | 42.211 | 40.930 | 46.607 | 45.853 | 45.146 |
| | (Sec.) | (0.001) | (0.001) | (0.001) | (0.001) | (0.001) | (0.001) | (0.001) | (0.001) | (0.001) |
| | $\vartheta(\varphi)$ | 37.590 | 34.254 | 29.236 | 43.467 | 42.164 | 40.884 | 46.559 | 45.804 | 45.096 |
| | (Min.) | (31.90) | (31.90) | (31.77) | (31.45) | (32.97) | (31.48) | (31.73) | (32.70) | (31.51) |
| | %Accuracy | 99.90 | 99.93 | 100.00 | 99.89 | 99.89 | 99.89 | 99.90 | 99.89 | 99.89 |
| 0.70 | $\ell(\varphi)$ | 22.750 | 20.717 | 17.792 | 26.538 | 25.674 | 24.839 | 28.651 | 28.126 | 27.640 |
| | (Sec.) | (0.001) | (0.001) | (0.001) | (0.001) | (0.001) | (0.001) | (0.001) | (0.001) | (0.001) |
| | $\vartheta(\varphi)$ | 22.734 | 20.708 | 17.796 | 26.514 | 25.650 | 24.818 | 28.626 | 28.102 | 27.615 |
| | (Min.) | (32.23) | (32.27) | (31.69) | (31.65) | (32.99) | (31.54) | (31.78) | (33.22) | (31.64) |
| | %Accuracy | 99.93 | 99.96 | 99.98 | 99.91 | 99.91 | 99.92 | 99.91 | 99.91 | 99.91 |
| 1.00 | $\ell(\varphi)$ | 13.295 | 12.259 | 10.858 | 15.382 | 14.888 | 14.421 | 16.641 | 16.322 | 16.030 |
| | (Sec.) | (0.001) | (0.001) | (0.001) | (0.001) | (0.001) | (0.001) | (0.001) | (0.001) | (0.001) |
| | $\vartheta(\varphi)$ | 13.289 | 12.255 | 10.861 | 15.372 | 14.877 | 14.411 | 16.630 | 16.310 | 16.018 |
| | (Min.) | (31.56) | (31.71) | (31.77) | (31.50) | (31.49) | (31.53) | (31.49) | (31.61) | (31.65) |
| | %Accuracy | 99.95 | 99.97 | 99.97 | 99.93 | 99.93 | 99.93 | 99.93 | 99.93 | 99.93 |
| 1.50 | $\ell(\varphi)$ | 7.543 | 7.140 | 6.651 | 8.460 | 8.230 | 8.020 | 9.077 | 8.916 | 8.771 |
| | (Sec.) | (0.001) | (0.001) | (0.001) | (0.001) | (0.001) | (0.001) | (0.001) | (0.001) | (0.001) |
| | $\vartheta(\varphi)$ | 7.541 | 7.139 | 6.652 | 8.456 | 8.227 | 8.017 | 9.072 | 8.911 | 8.767 |
| | (Min.) | (31.97) | (31.53) | (31.52) | (31.47) | (31.49) | (31.72) | (31.72) | (31.69) | (31.82) |

(*Continued*)

**Table 2.** (Continued)

| $\delta$ | $\eta$ | 3.0 | | | 3.5 | | | 4.0 | | |
|---|---|---|---|---|---|---|---|---|---|---|
| | $b$ | 4.300562 | 4.65323 | 5.091801 | 3.498061 | 3.70427 | 3.888005 | 2.884593 | 3.056922 | 3.20299 |
| | Model | [1] | [2] | [3] | [1] | [2] | [3] | [1] | [2] | [3] |
| | %Accuracy | 99.97 | 99.99 | 99.98 | 99.95 | 99.96 | 99.96 | 99.94 | 99.94 | 99.95 |
| 2.0 | $\ell(\varphi)$ | 5.296 | 5.115 | 4.924 | 5.765 | 5.641 | 5.531 | 6.112 | 6.020 | 5.937 |
| | (Sec.) | (0.001) | (0.001) | (0.001) | (0.001) | (0.001) | (0.001) | (0.001) | (0.001) | (0.001) |
| | $\vartheta(\varphi)$ | 5.295 | 5.114 | 4.924 | 5.763 | 5.639 | 5.529 | 6.110 | 6.017 | 5.935 |
| | (Min.) | (31.58) | (31.90) | (32.02) | (31.51) | (31.63) | (31.89) | (31.59) | (31.59) | (31.93) |
| | %Accuracy | 99.78 | 99.78 | 99.80 | 99.80 | 99.79 | 99.79 | 99.83 | 99.82 | 99.81 |

The results are expressed as percentage accuracy with computational times in parentheses for the analytical IE method (seconds) and the numerical IE method (minutes).

[1] SARFIMAX(2, 0.15, 1, 1)$_{12}$, [2] SARFIMAX(2, 0.30, 1, 1)$_{12}$ and [3] SARFIMAX(1, 0.45, 1, 1)$_{12}$

**Table 3. Comparison of the ARL$_1$ values obtained using the analytical and numerical ARL methods for long-memory SARFIMAX(P, D, Q, r)$_s$ processes with exponential white noise on a CUSUM control chart for ARL$_0$ = 370 with $\Phi_1 = -0.1$, $\Phi_2 = 0.2$, $\Theta_1 = 0.1$, and $\omega_1 = 0.5$.**

| $\delta$ | $\eta$ | 3.0 | | | 3.5 | | | 4.0 | | |
|---|---|---|---|---|---|---|---|---|---|---|
| | $b$ | 4.021805 | 4.387763 | 4.80233 | 3.304131 | 3.553436 | 3.776275 | 2.71586 | 2.93165 | 3.115083 |
| | Model | [−1] | [−2] | [−3] | [−1] | [−2] | [−3] | [−1] | [−2] | [−3] |
| 0.01 | $\ell(\varphi)$ | 346.040 | 344.922 | 343.186 | 347.482 | 347.072 | 346.628 | 348.162 | 347.953 | 347.741 |
| | (Sec.) | (0.001) | (0.001) | (0.001) | (0.001) | (0.001) | (0.001) | (0.001) | (0.001) | (0.001) |
| | $\vartheta(\varphi)$ | 345.294 | 344.164 | 342.454 | 346.820 | 346.375 | 345.904 | 347.599 | 347.352 | 347.109 |
| | (Min.) | (31.80) | (31.56) | (31.88) | (32.35) | (31.53) | (31.91) | (31.91) | (31.61) | (32.01) |
| | %Accuracy | 99.78 | 99.78 | 99.79 | 99.81 | 99.80 | 99.79 | 99.84 | 99.83 | 99.82 |
| 0.03 | $\ell(\varphi)$ | 303.857 | 301.000 | 296.592 | 307.581 | 306.518 | 305.372 | 309.351 | 308.807 | 308.253 |
| | (Sec.) | (0.001) | (0.001) | (0.001) | (0.001) | (0.001) | (0.001) | (0.001) | (0.001) | (0.001) |
| | $\vartheta(\varphi)$ | 303.223 | 300.363 | 295.993 | 307.010 | 305.918 | 304.752 | 308.863 | 308.286 | 307.706 |
| | (Min.) | (31.95) | (31.79) | (31.73) | (32.08) | (31.57) | (31.45) | (31.70) | (31.91) | (31.56) |
| | %Accuracy | 99.79 | 99.79 | 99.80 | 99.81 | 99.80 | 99.80 | 99.84 | 99.83 | 99.82 |
| 0.05 | $\ell(\varphi)$ | 268.149 | 264.067 | 257.818 | 273.512 | 271.975 | 270.323 | 276.085 | 275.292 | 274.486 |
| | (Sec.) | (0.001) | (0.001) | (0.001) | (0.001) | (0.001) | (0.001) | (0.001) | (0.001) | (0.001) |
| | $\vartheta(\varphi)$ | 267.606 | 263.529 | 257.325 | 273.016 | 271.457 | 269.790 | 275.659 | 274.838 | 274.011 |
| | (Min.) | (31.69) | (31.83) | (31.67) | (31.93) | (31.67) | (31.60) | (32.13) | (31.79) | (31.74) |
| | %Accuracy | 99.80 | 99.80 | 99.81 | 99.82 | 99.81 | 99.80 | 99.85 | 99.84 | 99.83 |
| 0.10 | $\ell(\varphi)$ | 200.161 | 194.503 | 185.987 | 207.750 | 205.554 | 203.214 | 211.473 | 210.317 | 209.151 |
| | (Sec.) | (0.001) | (0.001) | (0.001) | (0.001) | (0.001) | (0.001) | (0.001) | (0.001) | (0.001) |
| | $\vartheta(\varphi)$ | 199.786 | 194.142 | 185.676 | 207.396 | 205.187 | 202.840 | 211.164 | 209.990 | 208.809 |
| | (Min.) | (32.07) | (31.65) | (31.57) | (31.93) | (31.47) | (31.43) | (31.68) | (31.52) | (31.45) |
| | %Accuracy | 99.81 | 99.81 | 99.83 | 99.83 | 99.82 | 99.82 | 99.85 | 99.84 | 99.84 |
| 0.30 | $\ell(\varphi)$ | 78.327 | 73.410 | 66.434 | 85.402 | 83.288 | 81.098 | 89.151 | 87.962 | 86.786 |
| | (Sec.) | (0.001) | (0.001) | (0.001) | (0.001) | (0.001) | (0.001) | (0.001) | (0.001) | (0.001) |
| | $\vartheta(\varphi)$ | 78.217 | 73.315 | 66.371 | 85.286 | 83.172 | 80.983 | 89.045 | 87.851 | 86.672 |
| | (Min.) | (31.79) | (31.52) | (31.62) | (31.65) | (31.40) | (31.61) | (32.02) | (31.58) | (31.52) |
| | %Accuracy | 99.86 | 99.87 | 99.91 | 99.86 | 99.86 | 99.86 | 99.88 | 99.87 | 99.87 |
| 0.50 | $\ell(\varphi)$ | 39.925 | 36.847 | 32.690 | 44.618 | 43.179 | 41.722 | 47.263 | 46.410 | 45.579 |
| | (Sec.) | (0.001) | (0.001) | (0.001) | (0.001) | (0.001) | (0.001) | (0.001) | (0.001) | (0.001) |
| | $\vartheta(\varphi)$ | 39.882 | 36.812 | 32.672 | 44.569 | 43.130 | 41.675 | 47.217 | 46.361 | 45.530 |
| | (Min.) | (32.24) | (31.93) | (31.69) | (32.34) | (31.56) | (31.60) | (32.75) | (31.58) | (31.55) |

(*Continued*)

**Table 3.** (*Continued*)

| $\delta$ | $\eta$ | 3.0 | | | 3.5 | | | 4.0 | | |
|---|---|---|---|---|---|---|---|---|---|---|
| | $b$ | 4.021805 | 4.387763 | 4.80233 | 3.304131 | 3.553436 | 3.776275 | 2.71586 | 2.93165 | 3.115083 |
| | Model | [−1] | [−2] | [−3] | [−1] | [−2] | [−3] | [−1] | [−2] | [−3] |
| | %Accuracy | 99.89 | 99.91 | 99.94 | 99.89 | 99.89 | 99.89 | 99.90 | 99.89 | 99.89 |
| 0.70 | $\ell(\varphi)$ | 24.195 | 22.270 | 19.779 | 27.280 | 26.313 | 25.354 | 29.113 | 28.513 | 27.937 |
| | (Sec.) | (0.001) | (0.001) | (0.001) | (0.001) | (0.001) | (0.001) | (0.001) | (0.001) | (0.001) |
| | $\vartheta(\varphi)$ | 24.174 | 22.255 | 19.774 | 27.255 | 26.289 | 25.331 | 29.089 | 28.488 | 27.913 |
| | (Min.) | (31.93) | (31.75) | (31.71) | (33.06) | (31.70) | (31.73) | (33.29) | (31.74) | (31.61) |
| | %Accuracy | 99.91 | 99.93 | 99.97 | 99.91 | 99.91 | 99.91 | 99.92 | 99.91 | 99.91 |
| 1.00 | $\ell(\varphi)$ | 14.067 | 13.045 | 11.799 | 15.816 | 15.253 | 14.707 | 16.926 | 16.556 | 16.208 |
| | (Sec.) | (0.001) | (0.001) | (0.001) | (0.001) | (0.001) | (0.001) | (0.001) | (0.001) | (0.001) |
| | $\vartheta(\varphi)$ | 14.058 | 13.039 | 11.797 | 15.805 | 15.242 | 14.697 | 16.915 | 16.545 | 16.196 |
| | (Min.) | (31.93) | (32.00) | (31.83) | (32.00) | (31.76) | (31.52) | (32.03) | (32.02) | (31.58) |
| | %Accuracy | 99.94 | 99.95 | 99.98 | 99.93 | 99.93 | 99.93 | 99.94 | 99.93 | 99.93 |
| 1.50 | $\ell(\varphi)$ | 7.866 | 7.442 | 6.972 | 8.667 | 8.399 | 8.149 | 9.223 | 9.034 | 8.859 |
| | (Sec.) | (0.001) | (0.001) | (0.001) | (0.001) | (0.001) | (0.001) | (0.001) | (0.001) | (0.001) |
| | $\vartheta(\varphi)$ | 7.863 | 7.440 | 6.972 | 8.663 | 8.395 | 8.145 | 9.219 | 9.029 | 8.854 |
| | (Min.) | (31.82) | (31.80) | (31.60) | (32.07) | (31.82) | (31.58) | (32.05) | (31.86) | (31.74) |
| | %Accuracy | 99.96 | 99.97 | 100.00 | 99.95 | 99.95 | 99.95 | 99.96 | 99.94 | 99.94 |
| 2.0 | $\ell(\varphi)$ | 5.453 | 5.249 | 5.046 | 5.879 | 5.731 | 5.598 | 6.198 | 6.088 | 5.987 |
| | (Sec.) | (0.001) | (0.001) | (0.001) | (0.001) | (0.001) | (0.001) | (0.001) | (0.001) | (0.001) |
| | $\vartheta(\varphi)$ | 5.451 | 5.248 | 5.046 | 5.877 | 5.730 | 5.596 | 6.196 | 6.085 | 5.985 |
| | (Min.) | (31.67) | (31.69) | (31.73) | (32.97) | (31.59) | (31.45) | (33.21) | (31.57) | (31.61) |
| | %Accuracy | 99.78 | 99.78 | 99.79 | 99.81 | 99.80 | 99.79 | 99.84 | 99.83 | 99.82 |

The results are expressed as percentage accuracy with computational times in parentheses for the analytical IE method (seconds) and the numerical IE method (minutes).

[−1] SARFIMAX(2, 0.15, 1, 1)$_{12}$, [−2] SARFIMAX(2, 0.30, 1, 1)$_{12}$ and [−3] SARFIMAX(1, 0.45, 1, 1)$_{12}$

**Table 4. Comparison of the ARL$_1$ values obtained using the analytical and numerical ARL methods for long-memory SARFIMAX(*P*, *D*, *Q*, *r*)$_s$ processes with exponential white noise on a CUSUM control chart for ARL$_0$ = 500 with $\Phi_1 = 0.1$, $\Phi_2 = 0.2$, $\Theta_1 = 0.1$, and $\omega_1 = 0.5$.**

| $\delta$ | $\eta$ | 3.0 | | | 3.5 | | | 4.0 | | |
|---|---|---|---|---|---|---|---|---|---|---|
| | $b$ | 4.677214 | 5.08063 | 5.698833 | 3.82688 | 4.04085 | 4.233519 | 3.19902 | 3.37441 | 3.52359 |
| | Model | [1] | [2] | [3] | [1] | [2] | [3] | [1] | [2] | [3] |
| 0.01 | $\ell(\varphi)$ | 464.199 | 461.718 | 455.617 | 467.408 | 466.785 | 466.133 | 468.690 | 468.395 | 468.107 |
| | (Sec.) | (0.001) | (0.001) | (0.001) | (0.001) | (0.001) | (0.001) | (0.001) | (0.001) | (0.001) |
| | $\vartheta(\varphi)$ | 463.088 | 460.632 | 454.740 | 466.388 | 465.729 | 465.052 | 467.803 | 467.468 | 467.147 |
| | (Min.) | (33.10) | (32.63) | (32.83) | (33.07) | (32.62) | (33.01) | (33.31) | (32.94) | (33.00) |
| | %Accuracy | 99.76 | 99.76 | 99.81 | 99.78 | 99.77 | 99.77 | 99.81 | 99.80 | 99.79 |
| 0.03 | $\ell(\varphi)$ | 401.849 | 395.580 | 380.336 | 410.025 | 408.439 | 406.773 | 413.348 | 412.586 | 411.842 |
| | (Sec.) | (0.001) | (0.001) | (0.001) | (0.001) | (0.001) | (0.001) | (0.001) | (0.001) | (0.001) |
| | $\vartheta(\varphi)$ | 400.926 | 394.702 | 379.704 | 409.155 | 407.542 | 405.859 | 412.584 | 411.789 | 411.019 |
| | (Min.) | (33.14) | (32.73) | (33.09) | (32.87) | (32.47) | (32.62) | (33.33) | (33.00) | (33.01) |
| | %Accuracy | 99.77 | 99.78 | 99.83 | 99.79 | 99.78 | 99.78 | 99.82 | 99.81 | 99.80 |

(*Continued*)

**Table 4.** (Continued)

| $\delta$ | $\eta$ | 3.0 | | | 3.5 | | | 4.0 | | |
|---|---|---|---|---|---|---|---|---|---|---|
| | $b$ | 4.677214 | 5.08063 | 5.698833 | 3.82688 | 4.04085 | 4.233519 | 3.19902 | 3.37441 | 3.52359 |
| | **Model** | [1] | [2] | [3] | [1] | [2] | [3] | [1] | [2] | [3] |
| 0.05 | $\ell(\varphi)$ | 349.815 | 340.971 | 319.685 | 361.464 | 359.193 | 356.813 | 366.257 | 365.156 | 364.082 |
| | (Sec.) | (0.001) | (0.001) | (0.001) | (0.001) | (0.001) | (0.001) | (0.001) | (0.001) | (0.001) |
| | $\vartheta(\varphi)$ | 349.043 | 340.257 | 319.234 | 360.717 | 358.427 | 356.035 | 365.596 | 364.467 | 363.372 |
| | (Min.) | (34.88) | (33.10) | (33.10) | (33.48) | (33.72) | (32.70) | (33.54) | (32.68) | (32.60) |
| | %Accuracy | 99.78 | 99.79 | 99.86 | 99.79 | 99.79 | 99.78 | 99.82 | 99.81 | 99.80 |
| 0.10 | $\ell(\varphi)$ | 252.977 | 241.068 | 213.082 | 269.050 | 265.877 | 262.570 | 275.852 | 274.279 | 272.752 |
| | (Sec.) | (0.001) | (0.001) | (0.001) | (0.001) | (0.001) | (0.001) | (0.001) | (0.001) | (0.001) |
| | $\vartheta(\varphi)$ | 252.471 | 240.631 | 212.901 | 268.529 | 265.348 | 262.040 | 275.382 | 273.791 | 272.251 |
| | (Min.) | (33.53) | (32.76) | (32.82) | (33.34) | (32.73) | (32.77) | (33.60) | (32.84) | (33.03) |
| | %Accuracy | 99.80 | 99.82 | 99.92 | 99.81 | 99.80 | 99.80 | 99.83 | 99.82 | 99.82 |
| 0.30 | $\ell(\varphi)$ | 89.915 | 80.523 | 60.164 | 103.720 | 100.865 | 97.964 | 110.139 | 108.614 | 107.159 |
| | (Sec.) | (0.001) | (0.001) | (0.001) | (0.001) | (0.001) | (0.001) | (0.001) | (0.001) | (0.001) |
| | $\vartheta(\varphi)$ | 89.792 | 80.444 | 60.213 | 103.563 | 100.711 | 97.816 | 109.988 | 108.459 | 107.003 |
| | (Min.) | (34.90) | (32.97) | (32.91) | (33.57) | (32.55) | (32.76) | (33.91) | (32.83) | (32.98) |
| | %Accuracy | 99.86 | 99.90 | 99.92 | 99.85 | 99.85 | 99.85 | 99.86 | 99.86 | 99.85 |
| 0.50 | $\ell(\varphi)$ | 43.245 | 37.792 | 26.715 | 51.847 | 50.000 | 48.162 | 56.161 | 55.113 | 54.128 |
| | (Sec.) | (0.001) | (0.001) | (0.001) | (0.001) | (0.001) | (0.001) | (0.001) | (0.001) | (0.001) |
| | $\vartheta(\varphi)$ | 43.204 | 37.774 | 26.757 | 51.784 | 49.94 | 48.107 | 56.097 | 55.049 | 54.064 |
| | (Min.) | (34.82) | (32.66) | (33.08) | (34.70) | (32.70) | (33.02) | (35.00) | (32.58) | (32.95) |
| | %Accuracy | 99.91 | 99.95 | 99.84 | 99.88 | 99.88 | 99.89 | 99.89 | 99.88 | 99.88 |
| 0.70 | $\ell(\varphi)$ | 25.368 | 22.161 | 15.994 | 30.748 | 29.556 | 28.391 | 33.623 | 32.912 | 32.251 |
| | (Sec.) | (0.001) | (0.001) | (0.001) | (0.001) | (0.001) | (0.001) | (0.001) | (0.001) | (0.001) |
| | $\vartheta(\varphi)$ | 25.352 | 22.156 | 16.025 | 30.718 | 29.528 | 28.365 | 33.591 | 32.88 | 32.220 |
| | (Min.) | (32.81) | (32.77) | (32.98) | (32.48) | (32.64) | (33.01) | (32.47) | (32.78) | (33.28) |
| | %Accuracy | 99.94 | 99.98 | 99.81 | 99.90 | 99.91 | 99.91 | 99.90 | 99.90 | 99.90 |
| 1.00 | $\ell(\varphi)$ | 14.439 | 12.861 | 10.048 | 17.311 | 19.649 | 16.016 | 18.976 | 18.555 | 18.169 |
| | (Sec.) | (0.001) | (0.001) | (0.001) | (0.001) | (0.001) | (0.001) | (0.001) | (0.001) | (0.001) |
| | $\vartheta(\varphi)$ | 14.433 | 12.853 | 10.041 | 17.299 | 16.637 | 16.006 | 18.962 | 18.541 | 18.156 |
| | (Min.) | (32.47) | (32.80) | (33.11) | (32.51) | (32.78) | (33.02) | (32.56) | (32.92) | (33.25) |
| | %Accuracy | 99.96 | 99.94 | 99.93 | 99.93 | 84.67 | 99.94 | 99.93 | 99.92 | 99.93 |
| 1.50 | $\ell(\varphi)$ | 8.033 | 7.444 | 6.515 | 9.250 | 8.953 | 8.678 | 10.040 | 9.835 | 9.650 |
| | (Sec.) | (0.001) | (0.001) | (0.001) | (0.001) | (0.001) | (0.001) | (0.001) | (0.001) | (0.001) |
| | $\vartheta(\varphi)$ | 8.031 | 7.435 | 6.520 | 9.246 | 8.949 | 8.675 | 10.035 | 9.83 | 9.645 |
| | (Min.) | (37.45) | (33.08) | (32.76) | (37.73) | (32.96) | (32.46) | (37.45) | (32.91) | (32.74) |
| | %Accuracy | 99.98 | 99.88 | 99.92 | 99.96 | 99.96 | 99.97 | 99.95 | 99.95 | 99.95 |
| 2.0 | $\ell(\varphi)$ | 5.593 | 5.337 | 4.989 | 6.201 | 6.045 | 5.904 | 6.637 | 6.521 | 6.418 |
| | (Sec.) | (0.001) | (0.001) | (0.001) | (0.001) | (0.001) | (0.001) | (0.001) | (0.001) | (0.001) |
| | $\vartheta(\varphi)$ | 5.592 | 5.337 | 4.991 | 6.199 | 6.043 | 5.903 | 6.635 | 6.518 | 6.416 |
| | (Min.) | (36.97) | (32.85) | (32.92) | (37.36) | (32.87) | (32.62) | (37.61) | (32.86) | (32.90) |
| | %Accuracy | 99.76 | 99.76 | 99.81 | 99.78 | 99.77 | 99.77 | 99.81 | 99.80 | 99.79 |

The results are expressed as percentage accuracy with computational times in parentheses for the analytical IE method (seconds) and the numerical IE method (minutes).

[1] SARFIMAX(2, 0.15, 1, 1)$_{12}$, [2] SARFIMAX(2, 0.30, 1, 1)$_{12}$ and [3] SARFIMAX(1, 0.45, 1, 1)$_{12}$

**Table 5. Comparison of the ARL$_1$ values obtained using the analytical and numerical ARL methods for long-memory SARFIMAX(*P*, *D*, *Q*, *r*)$_s$ processes with exponential white noise on a CUSUM control chart for ARL$_0$ = 500 with $\Phi_1 = -0.1$, $\Phi_2 = 0.2$, $\Theta_1 = 0.1$, and $\omega_1 = 0.5$.**

| $\delta$ | $\eta$ | 3.0 | | | 3.5 | | | 4.0 | | |
|---|---|---|---|---|---|---|---|---|---|---|
| | $b$ | 4.375410 | 4.774041 | 5.266000 | 3.627221 | 3.884130 | 4.116095 | 3.027800 | 3.246850 | 3.433746 |
| | Model | [−1] | [−2] | [−3] | [−1] | [−2] | [−3] | [−1] | [−2] | [−3] |
| 0.01 | $\ell(\varphi)$ | 465.589 | 463.677 | 460.320 | 467.884 | 467.245 | 466.541 | 468.932 | 468.612 | 468.284 |
| | (Sec.) | (0.001) | (0.001) | (0.001) | (0.001) | (0.001) | (0.001) | (0.001) | (0.001) | (0.001) |
| | $\vartheta(\varphi)$ | 464.493 | 462.566 | 459.271 | 466.903 | 466.215 | 465.474 | 468.086 | 467.714 | 467.344 |
| | (Min.) | (33.38) | (33.31) | (34.02) | (33.45) | (33.11) | (33.96) | (33.70) | (33.20) | (33.98) |
| | %Accuracy | 99.76 | 99.76 | 99.77 | 99.79 | 99.78 | 99.77 | 99.82 | 99.81 | 99.80 |
| 0.03 | $\ell(\varphi)$ | 405.380 | 400.527 | 391.952 | 411.268 | 409.621 | 407.816 | 413.982 | 413.149 | 412.301 |
| | (Sec.) | (0.001) | (0.001) | (0.001) | (0.001) | (0.001) | (0.001) | (0.001) | (0.001) | (0.001) |
| | $\vartheta(\varphi)$ | 404.458 | 399.609 | 391.121 | 410.428 | 408.743 | 406.912 | 413.252 | 412.377 | 411.493 |
| | (Min.) | (33.31) | (33.20) | (34.00) | (32.65) | (32.92) | (33.65) | (33.23) | (33.39) | (34.05) |
| | %Accuracy | 99.77 | 99.77 | 99.79 | 99.80 | 99.79 | 99.78 | 99.82 | 99.81 | 99.80 |
| 0.05 | $\ell(\varphi)$ | 354.826 | 347.946 | 335.848 | 363.256 | 360.888 | 358.304 | 367.177 | 365.971 | 364.744 |
| | (Sec.) | (0.001) | (0.001) | (0.001) | (0.001) | (0.001) | (0.001) | (0.001) | (0.001) | (0.001) |
| | $\vartheta(\varphi)$ | 354.046 | 347.181 | 335.186 | 362.532 | 360.135 | 357.532 | 366.544 | 365.302 | 364.047 |
| | (Min.) | (32.95) | (32.58) | (33.43) | (32.61) | (32.50) | (34.70) | (32.90) | (32.48) | (33.56) |
| | %Accuracy | 99.78 | 99.78 | 99.80 | 99.80 | 99.79 | 99.78 | 99.83 | 99.82 | 99.81 |
| 0.10 | $\ell(\varphi)$ | 259.828 | 250.440 | 234.225 | 271.581 | 268.246 | 264.639 | 277.176 | 275.444 | 273.693 |
| | (Sec.) | (0.001) | (0.001) | (0.001) | (0.001) | (0.001) | (0.001) | (0.001) | (0.001) | (0.001) |
| | $\vartheta(\varphi)$ | 259.301 | 249.946 | 233.842 | 271.072 | 267.722 | 264.109 | 276.725 | 274.969 | 273.199 |
| | (Min.) | (32.77) | (32.62) | (33.38) | (32.90) | (32.51) | (33.56) | (32.95) | (32.58) | (33.59) |
| | %Accuracy | 99.80 | 99.80 | 99.84 | 99.81 | 99.80 | 99.80 | 99.84 | 99.83 | 99.82 |
| 0.30 | $\ell(\varphi)$ | 95.610 | 87.865 | 75.343 | 106.059 | 102.991 | 99.771 | 111.449 | 109.742 | 108.054 |
| | (Sec.) | (0.001) | (0.001) | (0.001) | (0.001) | (0.001) | (0.001) | (0.001) | (0.001) | (0.001) |
| | $\vartheta(\varphi)$ | 95.468 | 87.751 | 75.293 | 105.902 | 102.835 | 99.619 | 111.301 | 109.590 | 107.898 |
| | (Min.) | (34.60) | (32.70) | (33.70) | (34.04) | (32.38) | (33.44) | (34.33) | (32.77) | (33.56) |
| | %Accuracy | 99.85 | 99.87 | 99.93 | 99.85 | 99.85 | 99.85 | 99.87 | 99.86 | 99.86 |
| 0.50 | $\ell(\varphi)$ | 46.698 | 42.031 | 34.889 | 53.391 | 51.372 | 49.302 | 57.073 | 55.887 | 54.732 |
| | (Sec.) | (0.001) | (0.001) | (0.001) | (0.001) | (0.001) | (0.001) | (0.001) | (0.001) | (0.001) |
| | $\vartheta(\varphi)$ | 46.646 | 41.995 | 34.855 | 53.327 | 51.310 | 49.244 | 57.011 | 55.826 | 54.668 |
| | (Min.) | (34.46) | (35.26) | (34.11) | (34.03) | (32.48) | (33.65) | (34.29) | (32.74) | (33.73) |
| | %Accuracy | 99.89 | 99.91 | 99.90 | 99.88 | 99.88 | 99.88 | 99.89 | 99.89 | 99.88 |
| 0.70 | $\ell(\varphi)$ | 27.477 | 24.642 | 20.506 | 31.762 | 30.439 | 29.111 | 34.249 | 33.436 | 32.655 |
| | (Sec.) | (0.001) | (0.001) | (0.001) | (0.001) | (0.001) | (0.001) | (0.001) | (0.001) | (0.001) |
| | $\vartheta(\varphi)$ | 27.453 | 24.628 | 20.509 | 31.730 | 30.409 | 29.084 | 34.218 | 33.406 | 32.623 |
| | (Min.) | (34.20) | (32.67) | (33.34) | (34.15) | (32.47) | (33.45) | (34.21) | (32.90) | (33.45) |
| | %Accuracy | 99.91 | 99.94 | 99.99 | 99.90 | 99.90 | 99.91 | 99.91 | 99.91 | 99.90 |
| 1.00 | $\ell(\varphi)$ | 15.529 | 14.073 | 12.081 | 17.887 | 17.138 | 16.406 | 19.352 | 18.864 | 18.405 |
| | (Sec.) | (0.001) | (0.001) | (0.001) | (0.001) | (0.001) | (0.001) | (0.001) | (0.001) | (0.001) |
| | $\vartheta(\varphi)$ | 15.520 | 14.069 | 12.083 | 17.874 | 17.126 | 16.395 | 19.338 | 18.850 | 18.391 |
| | (Min.) | (33.38) | (33.10) | (33.61) | (32.76) | (32.80) | (33.72) | (33.83) | (32.84) | (34.34) |
| | %Accuracy | 99.94 | 99.97 | 99.98 | 99.93 | 99.93 | 99.93 | 99.93 | 99.93 | 99.92 |
| 1.50 | $\ell(\varphi)$ | 8.473 | 7.891 | 7.173 | 9.517 | 9.172 | 8.846 | 10.228 | 9.985 | 9.762 |
| | (Sec.) | (0.001) | (0.001) | (0.001) | (0.001) | (0.001) | (0.001) | (0.001) | (0.001) | (0.001) |
| | $\vartheta(\varphi)$ | 8.470 | 7.890 | 7.174 | 9.512 | 9.168 | 8.843 | 10.223 | 9.980 | 9.757 |
| | (Min.) | (33.52) | (34.36) | (33.16) | (32.95) | (33.34) | (32.61) | (33.02) | (33.67) | (32.94) |

(*Continued*)

**Table 5.** (Continued)

| $\delta$ | $\eta$ | 3.0 | | | 3.5 | | | 4.0 | | |
|---|---|---|---|---|---|---|---|---|---|---|
| | $b$ | 4.375410 | 4.774041 | 5.266000 | 3.627221 | 3.884130 | 4.116095 | 3.027800 | 3.246850 | 3.433746 |
| | Model | [−1] | [−2] | [−3] | [−1] | [−2] | [−3] | [−1] | [−2] | [−3] |
| | %Accuracy | 99.96 | 99.99 | 99.99 | 99.95 | 99.96 | 99.97 | 99.95 | 99.95 | 99.95 |
| 2.0 | $\ell(\varphi)$ | 5.802 | 5.529 | 5.228 | 6.345 | 6.159 | 5.990 | 6.745 | 6.606 | 6.480 |
| | (Sec.) | (0.001) | (0.001) | (0.001) | (0.001) | (0.001) | (0.001) | (0.001) | (0.001) | (0.001) |
| | $\vartheta(\varphi)$ | 5.801 | 5.528 | 5.229 | 6.343 | 6.157 | 5.988 | 6.742 | 6.604 | 6.478 |
| | (Min.) | (32.79) | (34.17) | (32.53) | (32.77) | (33.69) | (32.46) | (32.80) | (33.71) | (32.69) |
| | %Accuracy | 99.76 | 99.76 | 99.77 | 99.79 | 99.78 | 99.77 | 99.82 | 99.81 | 99.80 |

The results are expressed as percentage accuracy with computational times in parentheses for the analytical IE method (seconds) and the numerical IE method (minutes).

[−1] SARFIMAX(2, 0.15, 1, 1)$_{12}$, [−2] SARFIMAX(2, 0.30, 1, 1)$_{12}$ and [−3] SARFIMAX(1, 0.45, 1, 1)$_{12}$

**Table 6. The SARFIMAX(P, Q, D, r)$_{12}$ coefficients for the PTT stock price dataset from January 2003 to August 2018.**

| Variable | Coefficient | Std. Error | t-Statistic | P-value |
|---|---|---|---|---|
| D(PTT, 0, 12) | 0.496151 | 0.017080 | 29.04865 | 0.0000 |
| EUR | -0.114685 | 0.054130 | -2.118690 | 0.0355 |
| SAR(12) | 1.000000 | 0.035469 | 28.19334 | 0.0000 |
| SMA(12) | 0.799986 | 0.000298 | 2682.302 | 0.0000 |

long-memory process models [1] to [3]. For instance, for prespecified ARL$_0$ = 370 on long-memory SARFIMAX(2, 0.15, 1, 1)$_{12}$ with coefficient values $\Phi_1 = \pm 0.1$, $\Phi_2 = 0.2$, $\Theta_1 = 0.1$, and $\omega_1 = 0.5$, it can be seen that for $\eta = 3$, b = 4.300562 and 4.021805, respectively. Only the ARL$_0$, b, and $\eta$ values are involved in the design of the CUSUM control chart (see Eq (15)).

The performance assessment of the analytical and numerical IE methods for determining the ARL for detecting shifts in the process means requires computing their ARL$_1$ values, as described earlier. Tables 2–5 provide the numerical results obtain from both methods in which we report their ARL$_1$ values, percentage accuracy, and computational times. We summarize our findings as follows:

1. The results of the analytical and numerical IE methods for detecting changes in the process mean tended to decrease when the shift size was increased in the order of long-memory process models [1] SARFIMAX(2, 0.15, 1, 1)$_{12}$, [2] SARFIMAX(2, 0.30, 1, 1)$_{12}$ and [3] SARFIMAX(1, 0.45, 1, 1)$_{12}$.

2. The ARL$_1$ values for [3] SARFIMAX(1, 0.45, 1, 1)$_{12}$ and [2] SARFIMAX(2, 0.30, 1, 1)$_{12}$ were slightly lower than for [1] SARFIMAX(2, 0.15, 1, 1)$_{12}$. With negative coefficients, the results were similar: [−3] SARFIMAX(1, 0.45, 1, 1)$_{12}$ and [−2] SARFIMAX(2, 0.30, 1, 1)$_{12}$ were slightly lower than those [−1] SARFIMAX(2, 0.15, 1, 1)$_{12}$ (see the results for ARL$_0$ = 370 in Tables 2 and 3 and 500 in Tables 4 and 5).

3. The reference value ($\eta$) was inversely proportional to the upper control limit and directly proportional to ARL$_1$ for both methods.

4. The percentage accuracy results were ~99% in all cases, meaning that the proposed analytical IE method was very accurate.

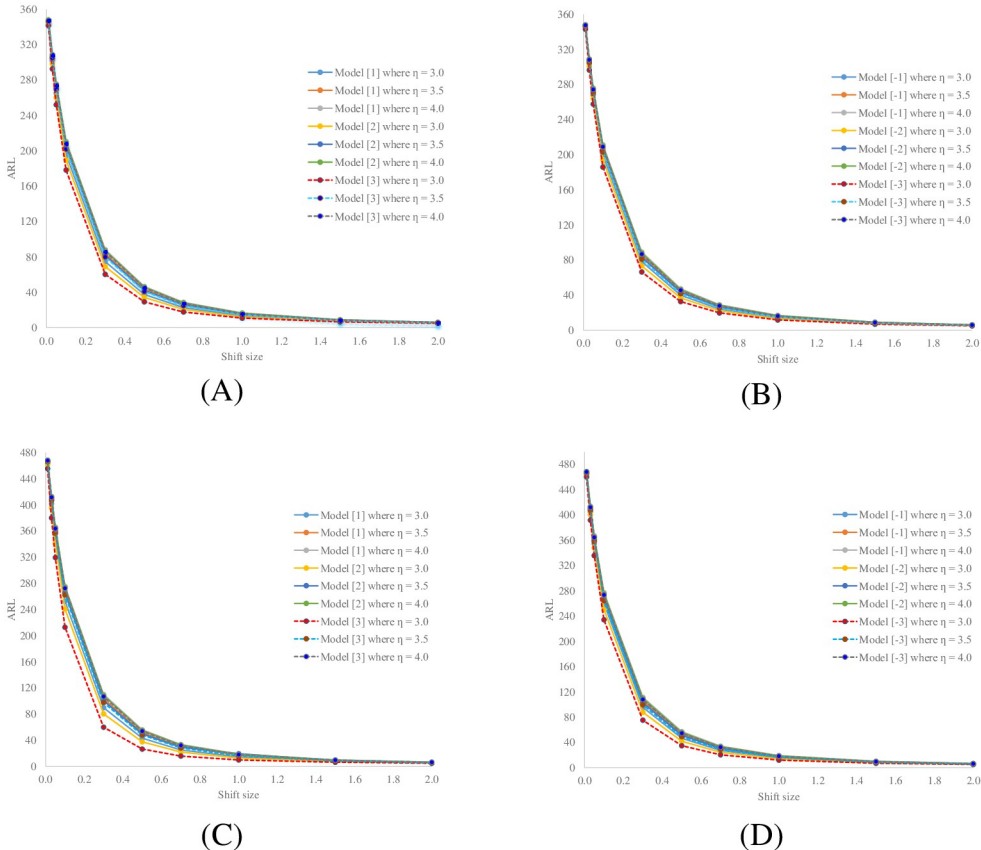

**Fig 1. Comparative performances of the analytical and numerical IEs for determining ARL$_1$ for detecting shifts in the mean of a long-memory process on a CUSUM control chart given reference value (*η*) = 3.0, 3.5 and 4.0.** (A) Models [1] to [3] for ARL$_0$ = 370, (B) models [−1] to [−3] for ARL$_0$ = 370, (C) models [1] to [3] for ARL$_0$ = 500, and (D) models [−1] to [−3] for ARL$_0$ = 500.

5. The values in parentheses in each column are the computational times for calculating ARL$_1$: 31–34 minutes for ARL$_0$ = 370 and 32–38 minutes for ARL$_0$ = 500 using the numerical IE but less than 1 second for both using the analytical IE.

Fig 1 shows plots of *δ* (shift size) versus ARL$_1$ value obtained by using the analytical IE for the various long-memory SARFIMAX(*P*, *D*, *Q*, *r*)$_s$ processes with different reference values (*η*) = 3.0, 3.5, 4.0 and for ARL$_0$ = 370 and 500. These graphs demonstrate the various characteristics of long-memory processes on the CUSUM control chart with coefficients $\Phi_1$ = 0.1, $\Phi_2$ = 0.2, $\Theta_1$ = 0.1, and $\omega_1$ = 0.5 in Fig 1A and 1C and $\Phi_1$ = −0.1, $\Phi_2$ = 0.2, $\Theta_1$ = 0.1, and $\omega_1$ = 0.5 in Fig 1B and 1D. It is evident that ARL$_1$ decreased rapidly for small shifts (0 < *δ* ≤ 0.05) and slightly less so for moderate shifts (0.05 < *δ* ≤ 0.20). In addition, the lowest reference parameter (*η*) = 3.0 was better for detecting small to moderate shifts in the process mean than 3.5 and 4. Moreover, when *D* was increased from 0.15 to 0.30, detecting changes was more sensitive. The results also show that the efficacy of both IEs for detecting changes was best for models [3] and [−3] with reference parameter (*η*) = 3.0 and = 370 and 500.

In summary, the results show that the analytical IE method is a good alternative to the numerical IE method for evaluating shifts in the process mean of a long-memory SARFIMAX (*P*, *D*, *Q*, *r*)$_s$ process with exponential white noise on a CUSUM control chart, especially because of the marked reduction in computational time.

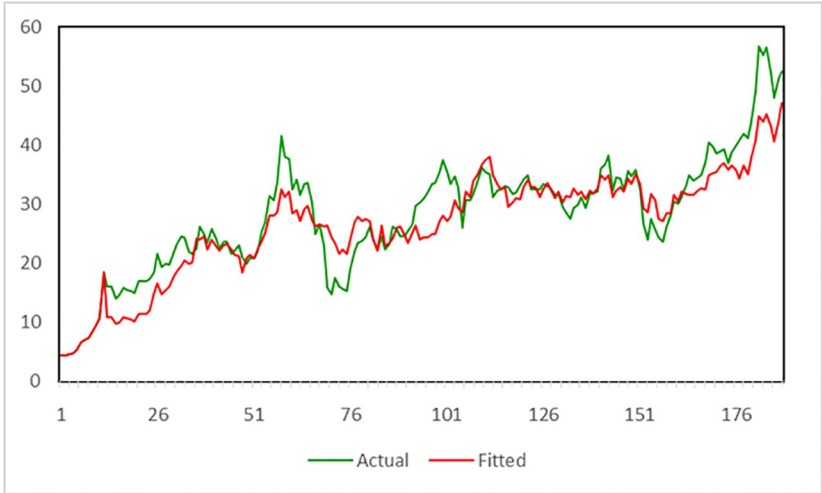

**Fig 2. The fitted SARFIMAX(1, 0.496151, 1 1)$_{12}$ model.**

## Illustrative example

In this section, the results of applying the proposed analytical and numerical IE methods for determining the ARL for monitoring changes in the mean of a long-memory SARFIMAX(*P, D, Q, r*)$_s$ processes with exponential white noise on a CUSUM control chart involving real data are provided. The real dataset comprises monthly stock price data for the PTT Public Company Ltd. in Thailand (https://th.investing.com/equities/ptt-historical-data) with the EUR/THB exchange rate as the exogenous variable (https://th.investing.com/currencies/eur-thb-historical-data). In this example, we used monthly data from January 2003 to August 2018 comprising 188 observations. A seasonal pattern exists when a series is influenced by a seasonal factor, which is a 12-month difference term in the model (*S* = 12). Accordingly, SARFIMAX(*P, D, Q, r*)$_{12}$ was selected as the model. Statistical software package Eviews 10 was utilized for SARFIMAX(*P, D, Q, r*)$_{12}$ filtering and estimating the model parameters.

The results in Table 6 reveal that the probability of variable *X* was significant (0.05; *p*-value = 0.0355), which indicates that the exchange rate does impact the stock price data for PTT. It can also be seen that the dataset is suitable for a long-memory SARFIMAX(1, 0.496151, 1 1)$_{12}$ model presenting statistically significant parameters with coefficients $\Phi_1$ = 1.00, $\Theta_1$ = 0.799986, $\omega_1$ = −0.114685 and the residual series behaving as white noise. The suitability of the model was checked by plotting the actual ($Y_t$) and fitted ($\hat{Y}_t$) components and appears to be in close agreement, as shown in Fig 2.

After that, the distribution of the white noise was confirmed as asymptotic exponential by using the Kolmogorov-Smirnov test. The results show that the white noise significantly fitted an exponential distribution (P-value = 0.091 > 0.05) with a mean of 3.42526 when the process was in-control, as reported in Table 7.

**Table 7. Testing the suitability of an exponential distribution for the exponential white noise.**

| Testing whether the white noise is exponentially distributed. Alternative hypothesis: two-sided | |
| --- | --- |
| One-sample Kolmogorov-Smirnov test | 1.24200 |
| Exponential parameter | 3.42526 |
| Asymp. Sig. (2-tailed) | 0.09100* |

**Table 8. Comparison of the ARL$_1$ values obtained via the analytical and numerical IE methods for detecting changes in the mean of the long-memory SARFIMAX (1, 0.496151, 1 1)$_{12}$ process on a CUSUM control chart for $\Phi_1$ = 1.00, $\Theta_1$ = 0.799986, and $\omega_1$ = −0.114685.**

| ARL$_0$ | ARL | $\delta$ | | | | | | | | | |
|---|---|---|---|---|---|---|---|---|---|---|---|
| | | 0.01 | 0.03 | 0.05 | 0.10 | 0.3 | 0.5 | 0.7 | 1.0 | 1.5 | 2.0 |
| 370 | $\ell(\varphi)$ | 361.218 | 344.416 | 328.571 | 292.746 | 190.263 | 129.346 | 91.510 | 58.135 | 31.553 | 19.744 |
| | (Sec.) | (0.001) | (0.001) | (0.001) | (0.001) | (0.001) | (0.001) | (0.001) | (0.001) | (0.001) | (0.001) |
| | $\vartheta(\varphi)$ | 360.476 | 343.724 | 327.926 | 292.203 | 189.987 | 129.205 | 91.439 | 58.113 | 31.557 | 19.753 |
| | (Min.) | (33.44) | (33.51) | (33.36) | (33.49) | (32.86) | (32.57) | (32.42) | (34.02) | (32.82) | (32.72) |
| | %Accuracy | 99.79 | 99.80 | 99.80 | 99.81 | 99.85 | 99.89 | 99.92 | 99.96 | 99.99 | 99.95 |
| 500 | $\ell(\varphi)$ | 486.626 | 461.144 | 437.247 | 383.728 | 235.376 | 151.703 | 102.263 | 61.059 | 30.789 | 18.538 |
| | (Sec.) | (0.001) | (0.001) | (0.001) | (0.001) | (0.001) | (0.001) | (0.001) | (0.001) | (0.001) | (0.001) |
| | $\vartheta(\varphi)$ | 485.592 | 460.198 | 436.382 | 383.037 | 235.104 | 151.615 | 102.256 | 61.095 | 30.833 | 18.573 |
| | (Min.) | (33.07) | (33.01) | (32.45) | (34.14) | (32.90) | (33.01) | (32.65) | (32.89) | (32.94) | (32.77) |
| | %Accuracy | 99.79 | 99.79 | 99.80 | 99.82 | 99.88 | 99.94 | 99.99 | 99.94 | 99.86 | 99.81 |

The results are expressed as percentage accuracy with the computational times in parentheses for the analytical IE (seconds) and numerical IE (minutes) methods.

Based on the results in Tables 6 and 7, the SARFIMAX(1, 0.496151, 1 1)$_{12}$ model ($Y_t$) comprised the following equation:

$$Y_t = Y_{t-12} - 0.496151 Y_{t-24} - 0.124999 Y_{t-36} - 0.06266 Y_{t-48} + 0.49615 Y_{t-12} + 0.12499 Y_{t-24} + 0.06266 Y_{t-36} + \xi_t - 0.799986 \xi_{t-12} - 0.114685 X_t,$$

where $\xi_t \sim Exp(\beta_0 = 3.42526)$.

Next, we calculated ARL$_0$ using the analytical and numerical IEs and chose the value of parameter ($\eta$) = 6.5 after calculating the control limits for ARL$_0$ = 370 and 500 using Eq (16) ($b$ = 17.37735 and 19.2241, respectively). The rationale behind the use of 370 and 500 of the in-control ARLs is that they give good results in practice. The performances of the two methods for determining the ARL for detecting shifts in the process mean of a long-memory SARFIMAX(1, 0.496151, 1 1)$_{12}$ on a CUSUM control chart are presented in Table 8 and Fig 3.

It can be seen that the ARL$_1$ results using the two methods are similar to those in Tables 2–5. The numerical results obtained from both methods were similar for both short- and long-term detections (ARL$_0$ = 370 and 500, respectively) for all shift sizes of the process mean. Once again, the computation time for the analytical IE was far less than that of the numerical IE

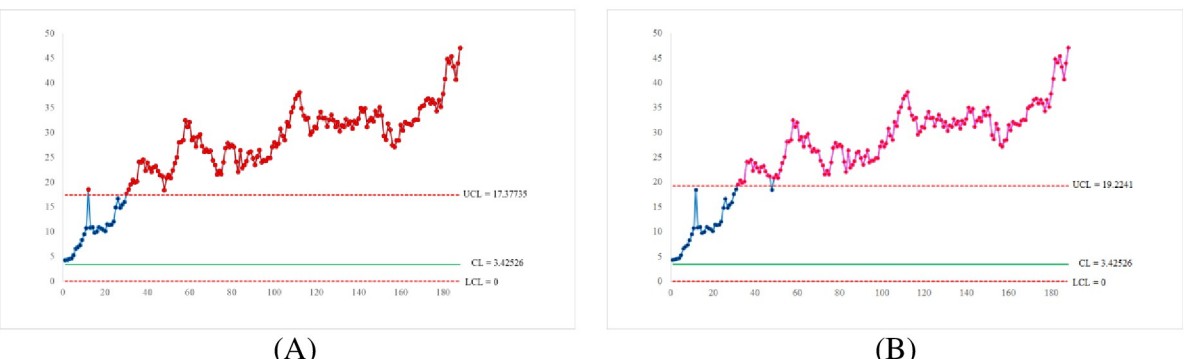

(A)                                              (B)

**Fig 3. Detecting shifts the mean on the long-memory SARFIMAX(1, 0.496151, 1 1)$_{12}$ process based on real data on an upper-side CUSUM control chart for (A) ARL$_0$ = 370 and (B) ARL$_0$ = 500.** CL, center line; UCL, upper control limit; LCL, lower control limit.

**Table 9. Comparison of the $ARL_1$ values obtained via the analytical IE on a CUSUM control chart and a numerical IE method on a EWMA control chart with $\lambda = 0.1$ for detecting changes in the mean of the long-memory SARFIMAX$(1, 0.496151, 1\ 1)_{12}$ process.**

| $ARL_0$ | $b$ | Control charts | $\delta$ | | | | | | | | | |
|---|---|---|---|---|---|---|---|---|---|---|---|---|
| | | | 0.01 | 0.03 | 0.05 | 0.10 | 0.3 | 0.5 | 0.7 | 1.0 | 1.5 | 2.0 |
| 370 | 17.37735 | CUSUM | 361.218 | 344.416 | 328.571 | 292.746 | 190.263 | **129.346** | **91.510** | **58.135** | **31.553** | **19.744** |
| | 0.2507646 | EWMA | **360.939** | **343.676** | **327.482** | **291.168** | **189.304** | 129.741 | 92.763 | 59.766 | 32.705 | 20.176 |
| 500 | 19.2241 | CUSUM | 486.626 | 461.144 | 437.247 | 383.728 | **235.376** | **151.703** | **102.263** | **61.059** | **30.789** | **18.538** |
| | 0.2725181 | EWMA | **486.394** | **460.637** | **436.678** | **383.657** | 240.346 | 160.496 | 112.564 | 71.008 | 37.950 | 23.040 |

method (less than 1 second versus approximately 33 minutes, respectively). This means that the analytical IE for determining $ARL_1$ is a good alternative for detecting changes in the process mean on a CUSUM control chart.

A SARFIMAX process involving real data was run on both CUSUM and EWMA control charts. The performances of the control charts were compared in terms of the ARL for detecting small to moderate shifts in the process mean; the ARL constructed using the analytical IE was used on the CUSUM control chart whereas the one using the numerical IE method was used on the EWMA control chart [28]. For the performance comparison, boundary values $b = 17.37735, 19.2241$ for the CUSUM control chart and $b = 0.2507646, 0.2725181$ for the EWMA control chart were used with prespecified $ARL_0 = 370$ or $ARL_0 = 500$, and smoothing parameter $\lambda$ for the EWMA control chart was determined as 0.1. The results of the comparison are summarized in Table 9. For $ARL_0 = 370$, the EWMA control chart provided a smaller $ARL_1$ than the CUSUM control chart for shift size $0 < \delta \leq 0.3$ whereas the CUSUM control chart provided a smaller $ARL_1$ than the EWMA control chart for shift size $0.5 < \delta \leq 2.0$. However, for $ARL_0 = 500$, the CUSUM control chart performed better than the EWMA control chart for shift sizes $0.3 < \delta \leq 2.0$. This means that the performance of the CUSUM control chart was more powerful than that of the EWMA chart when detecting moderate shifts in the process mean for $ARL_0 = 370$ and 500. Therefore, the CUSUM control chart was more efficacious than the EWMA control chart under these conditions, which is in accordance with the study results.

In Fig 3, the center line, the lower and upper control limits of the CUSUM control chart were CL = $\beta_0$, LCL = 0, UCL = 17.37735 and 19.2241for $ARL_0 = 370$ and 500, respectively. We can see that the signal was first given at the 12[th] and 32[nd]-47[th] time points for $ARL_0 = 370$ and 500, respectively (the observations in red plotted above the upper control limit). These results confirm the sensitivity of the proposed analytical IE for the ARL on a CUSUM control chart providing 176 and 156 out-of-control signals for $ARL_0 = 370$ and 500, respectively. This situation directly affects the stock price of PPT. Hence, tracking changes in exchange rates can be key both in terms of economy and finance. In an investor's view, the control chart above is the point in making profitable decisions for investors.

## Conclusions

Performance assessment of control chart can be measured by using the ARL determined via analytical and numerical IE methods used to analyze and approximate the ARL computation, respectively. We applied both methods to determine the ARL for monitoring changes in the mean of a long-memory SARFIMAX process with underlying exponential white noise on a CUSUM control chart. Computation of the ARL by using the numerical IE method was used to check the accuracy of the analytical IE method by using the percentage accuracy benchmark and comparing their computation times. The number of division points ($m$) for the numerical

IE method was large enough so that its ARL converged to that determined using the explicit formulas. The results show that the analytical IE was as accurate as the numerical IE method but required less computational time because its computational scheme was easier and could be completed in one stage.

The CUSUM control chart is suitable for detecting small and moderate parameter shifts for autocorrelated processes with bounded support [29, 30]. However, finding optimal parameters of the CUSUM control chart in these situations can be somewhat difficult. Therefore, obtaining the optimal values for the parameters of a CUSUM control chart should be studied further. The scope of the study could be extended in terms of the ARL and the methods used to compute it and could be applied to other types of processes including application to real data with other distributions of white noise such as gamma and Weibull from the exponential family.

## Supporting information

**S1 File.**
(DOCX)

**S1 Table.**
(DOCX)

## Author Contributions

**Conceptualization:** Wilasinee Peerajit.

**Data curation:** Yupaporn Areepong.

**Formal analysis:** Yupaporn Areepong, Wilasinee Peerajit.

**Funding acquisition:** Wilasinee Peerajit.

**Investigation:** Wilasinee Peerajit.

**Methodology:** Wilasinee Peerajit.

**Software:** Yupaporn Areepong.

**Validation:** Wilasinee Peerajit.

**Writing – original draft:** Wilasinee Peerajit.

**Writing – review & editing:** Wilasinee Peerajit.

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
