## [Decision Letter · Decision Letter 0]

29 Oct 2021

PONE-D-21-31542Integral Equation Solutions for the Average Run Length for Monitoring Shifts in the Mean of a Generalized Seasonal ARFIMAX(P, D, Q, r) Process Running on a CUSUM Control ChartPLOS ONE

Dear Dr. Peerajit,

Thank you for submitting your manuscript to PLOS ONE. After careful consideration, we feel that it has merit but does not fully meet PLOS ONE’s publication criteria as it currently stands. Therefore, we invite you to submit a revised version of the manuscript that addresses the points raised during the review process.

We look forward to receiving your revised manuscript.

Kind regards,

Maria Alessandra Ragusa, PhD Professor

Academic Editor

PLOS ONE

Journal Requirements:

"King Mongkut's University of Technology North Bangkok has received funding support from the National Science, Research and Innovation Fund (NSRF) (Grant No. KMUTNB-BasicR-64-20)."

"King Mongkut's University of Technology North Bangkok has received funding support from the National Science, Research and Innovation Fund (NSRF) (Grant No. KMUTNB-BasicR-64-20)."

Additional Editor Comments:

The paper, after minor revision according to the report, could be published. Then, make the suggested modifications and send back the new version.

Best regards.

Reviewers' comments:

Reviewer's Responses to Questions

**Comments to the Author**

1. Is the manuscript technically sound, and do the data support the conclusions?

Reviewer #1: Partly

2. Has the statistical analysis been performed appropriately and rigorously? 

Reviewer #1: Yes

3. Have the authors made all data underlying the findings in their manuscript fully available?

Reviewer #1: Yes

4. Is the manuscript presented in an intelligible fashion and written in standard English?

Reviewer #1: Yes

5. Review Comments to the Author

Reviewer #1: Comments for Manuscript Number: PONE-D-21-31542

Integral Equation Solutions for the Average Run Length for Monitoring Shifts in the

Mean of a Generalized Seasonal ARFIMAX(P, D, Q, r) Process Running on a CUSUM

Control Chart

Comment1:

What method do you apply to estimate numerical integral equation?. In abstract, you wrote Gauss-Legendre quadrature rules but in Definition 5: quadrature rules.

They are different.

Comment2:

repeat "IE"

Comment3:

Why do you use EUR/THB exchange rate to be the exogenous variable. Normally, Thai Baht is depended on the US dollar. You should also show the results of USD/THB as the exogenous variable.

Comment4:

Please insert the references about application of the time series to these fields.

Comment5:

It is wrong. Xi is an error term as a random variable. The error term cannot be written with equal sign in term of a pdf function. Please correct it.

Comment6:

Norm C(M) or infinity norm, please check.

Comment7:

You cannot prove the existence and uniqueness for the numerical integral equation. It is for the analytical integral equation.

6. PLOS authors have the option to publish the peer review history of their article (what does this mean?). If published, this will include your full peer review and any attached files.

Reviewer #1: No

---

## [Author Response · Author response to Decision Letter 0]

8 Jan 2022

Reviewer comments Revised manuscript

Comment 1: 

What method do you apply to estimate numerical integral equation?. In abstract, you wrote Gauss-Legendre quadrature rules but in Definition 5: quadrature rules. 

They are different.

The authors revised by reviewer suggestion.

Comment 2: 

repeat "IE" 

The authors revised by reviewer suggestion.

Comment 3: 

Why do you use EUR/THB exchange rate to be the exogenous variable. Normally, Thai Baht is depended on the US dollar. You should also show the results of USD/THB as the exogenous variable.

 The study of the effects of Thai stock indexes on the exchange rate as an exogenous variable. The relationship between the stock price index and exchange rate. It was found that the US dollar exchange rate against the baht and the Euro against the US dollar affected the Thai stock market index (see. Tsai, I-C. (2012)). A previous study mentions Thai stock indexes with the exchange rate of USD/THB as the exogenous variable. For instance, Sunthornwat R. and Areepong Y, 2020; Peerajit W., (article in press).

In this study, we investigated the relationship between Thai stocks and other interesting exchange rates, where EUR/THB is the exchange rate with the aforementioned relationship. Therefore, EUR/THB exchange rate to be the exogenous variable was applied.

Reference 

Tsai, I-Chun. The relationship between stock price index and exchange rate in Asian markets: A quantile regression approach. Journal of International Financial Markets Institutions and Money. 2012; 22(3), 609-621. 

Sunthornwat R, Areepong Y. Average Run Length on CUSUM Control Chart for Seasonal and Non-Seasonal Moving Average Processes with Exogenous Variables. Symmetry. 2020; 12(1), 173.

Peerajit W. Cumulative Sum Control Chart Applied to Monitor Shifts in the Mean of a Long-memory ARFIMAX (p, d*, q, r) Process with Exponential White Noise. The article in press of the Thailand Statistician Journal.

Comment4:

Please insert the references about application of the time series to these fields.

The authors revised by reviewer suggestion.

Comment5:

It is wrong. is an error term as a random variable. The error term cannot be written with equal sign in term of a pdf function. Please correct it. 

The authors revised by reviewer suggestion.

Comment6:

Norm C(M) or infinity norm, please check.

The authors revised by reviewer suggestion.

Comment7:

You cannot prove the existence and uniqueness for the numerical integral equation. It is for the analytical integral equation.

The authors revised by reviewer suggestion.

---

## [Editor Report · Decision Letter 1]

8 Feb 2022

Integral Equation Solutions for the Average Run Length for Monitoring Shifts in the Mean of a Generalized Seasonal ARFIMAX(P, D, Q, r) Process Running on a CUSUM Control Chart

PONE-D-21-31542R1

Dear Dr. Peerajit,

We’re pleased to inform you that your manuscript has been judged scientifically suitable for publication and will be formally accepted for publication once it meets all outstanding technical requirements.

Kind regards,

Maria Alessandra Ragusa, PhD Professor

Academic Editor

PLOS ONE

Additional Editor Comments (optional):

The revised version is now ready for publication.

Best regards.
---

## [Editor Report · Acceptance letter]

16 Feb 2022

PONE-D-21-31542R1 

Integral Equation Solutions for the Average Run Length for Monitoring Shifts in the Mean of a Generalized Seasonal Process Running on a CUSUM Control Chart 

Dear Dr. Peerajit:

I'm pleased to inform you that your manuscript has been deemed suitable for publication in PLOS ONE. Congratulations! Your manuscript is now with our production department. 

Kind regards, 

on behalf of

Dr. Maria Alessandra Ragusa 

Academic Editor

PLOS ONE